

# Exploring aerosol cloud interaction using VOCALS-REx aircraft measurements

Hailing Jia[1], Xiaoyan Ma[1] and Yangang Liu[2]

[1]Key Laboratory of Meteorological Disaster, Ministry of Education (KLME)/Joint International Research Laboratory of
Climate and Environment Change (ILCEC)/Collaborative Innovation Center on Forecast and Evaluation of Meteorological
Disasters (CIC-FEMD)/Key Laboratory for Aerosol-Cloud-Precipitation of China Meteorological Administration, Nanjing
University of Information Science & Technology, Nanjing 210044, China
[2]Environmental and Climate Sciences Department, Brookhaven National Laboratory, Upton, NY, USA

*Correspondence to*: Xiaoyan Ma (xma@nuist.edu.cn)

**Abstract.** In situ aircraft measurements during the VAMOS Ocean-Cloud-Atmosphere-Land Study-Regional Experiment

(VOCALS-REx) field campaign are employed to study the interaction between aerosol and stratocumulus over the southeast

Pacific Ocean, as well as entrainment process near the top of stratocumulus and its possible impacts on aerosol-cloud

interaction. Our analysis suggest that the increase of liquid water content (*LWC*) is mainly contributed by cloud droplet number

concentration ($N_d$) instead of effective radius of cloud droplets in the polluted case, in which more droplets form with smaller

size, while the opposite is true in the clean case. By looking into the influences of dynamical conditions and aerosol

microphysical properties on the cloud droplet formation, it is confirmed that cloud droplets are more easily to form under the

conditions with large vertical velocity and aerosol size. An increase in aerosol concentration tends to increase both $N_d$ and

relative dispersion ($\varepsilon$), while an increase in vertical velocity (*w*) often increases $N_d$ but decreases $\varepsilon$. After constraining the

differences of cloud dynamics, positive correlation between ε and $N_d$ become stronger, implying that perturbations of *w* could

weaken the influence of aerosol on ε, and hence may result in an underestimation of aerosol dispersion effect. The difference

of cloud microphysical properties between entrainment and non-entrainment zones confirms that the entrainment-mixing

mechanism is predominantly extreme inhomogeneous in the stratocumulus that capped by a sharp inversion, namely the

entrainment reduces $N_d$ and *LWC* by 28.9 % and 24.8 % on average, respectively, while the size of droplets is relatively

unaffected. In entrainment zone, smaller aerosols and drier air entrained from the top induce less cloud droplet with respect to

total in-cloud particles (0.56 ± 0.22) than the case in non-entrainment zone (0.73 ± 0.13) by inhibiting aerosol activation and

promoting cloud droplets evaporation.

## 1 Introduction

Stratocumulus plays a key role in the radiative energy budget of the Earth by reflecting incoming shortwave radiation

and thus cools the surface of the planet and offsets the warming by greenhouse gases (Hartmann et al., 1992). Stratocumulus





clouds are susceptible to aerosols, i.e. aerosol indirect effect (Twomey, 1974; Albrecht, 1989), which currently remains large

uncertainties (Lohmann and Feichter, 2005; Chen and Penner, 2005; Carslaw et al., 2013; McCoy et al., 2017).

      The marine stratocumulus overlaying the southeast Pacific Ocean (SEP) is the largest and most persistent clouds in the

world (Klein and Hartmann, 1993; Bretherton et al., 2004). Sources of anthropogenic aerosol from the Chilean and Peruvian

coasts, in contrast with relatively clean air masses from the Pacific Ocean, make the SEP an ideal region to explore the

interaction of aerosol and stratocumulus cloud topped boundary layers. The cloud properties from satellite retrievals exhibit a

gradient off the shore of Northern Chile. For example, cloud droplet number concentration decreased from 160 to 40 $cm^{-3}$

(George and Wood, 2010) and cloud droplet effective radius increased from 8 to 14 µm from the coast to about 1000 km

offshore (Wood et al., 2006). This gradient is plausibly attributed to anthropogenic aerosol near the coast. Huneeus et al. (2006)

found that during easterly wind events, sulfate increased one order of magnitude over SEP, which results in 1.6 to 2 fold

increase in cloud droplet number concentration. Based on observations from satellites and cruises, Wood et al. (2008)

suggested that open cellular convection within overcast stratocumulus is associated with reduced aerosol concentration, and

an air mass not passing through the Chilean coast, which further confirms the impact of aerosol on stratocumulus over SEP.

However, it is difficult to establish the generality of previous studies based on satellite remote sensing due to the absence of

in situ observations that provide vertical profiles of cloud and aerosol and detailed in-cloud processes.

The VAMOS (Variability of the American Monsoons) Ocean-Cloud-Atmosphere-Land Study-Regional Experiment

(VOCALS-REx), which includes multiple aircraft missions, ship and land-based measurements, took place in the region

extending from the near-coastal of northern Chile and southern Peru to the remote ocean in the SE Pacific during October–

November 2008 (Wood et al., 2011). Studies based on this field campaign provided more information about the properties of

aerosol, cloud and marine boundary layer over SEP. For instance, the multi-platform observations during VOCALS revealed

that the boundary layer was shallow and fairly well mixed near shore but deeper and decoupled offshore (Bretherton et al.,

2010). Twohy et al. (2013) found that higher aerosol concentrations near shore were associated with more but smaller cloud

droplets, less liquid water path (*LWP*), and thus attributed to a combined effect of anthropogenic aerosol and the physically

thinner clouds near shore. Nevertheless, an increase in *LWP* with the cloud condensation nuclei (CCN) concentrations was

found during the similar meteorological conditions (Zheng et al., 2010). Additionally, chemical components and sources of

aerosols during VOCALS-REx campaign have been discussed in several studies (Chand et al., 2010; Hawkins et al., 2010;

Allen et al., 2011; Twohy et al., 2013; Lee et al., 2014). Although these studies improved our understanding of aerosol, cloud

and boundary layer properties over SEP, the mechanisms of the detailed processes on interaction between aerosol and

stratocumulus cloud is still unclear.

      By employing in situ aircraft data collected by CIRPAS Twin Otter aircraft during VOCALS-REx, we investigate the

following issues in this study: (a) the relationships between aerosol and cloud properties; (b) cloud droplet formation and its





influencing factors; (c) dispersion effect (i.e., the influence of aerosol on the shape of cloud droplet size spectrum), and (d) entrainment process near the top of stratocumulus and its impact on cloud. This paper is organized as the follows: The instruments and measurement data are described in Sect. 2, and the main results are discussed in Sect. 3. A summary and discussion is given in Sect. 4.

## 2 Data and method

### 2.1 Aircraft Data

The Twin Otter operated by the Center for Interdisciplinary Remotely Piloted Aircraft Studies (CIRPAS) was aimed to observe aerosol, cloud microphysics, and turbulence near Point Alpha (20° S, 72° W) off the coast of Northern Chile from 16 October to 13 November 2008. A total of 19 flights were carried out, each of which conducting about 3 hours of sampling at Point Alpha and including several soundings and horizontal legs near the ocean surface, below the cloud, near the cloud base, within the cloud, near the cloud top, and above the cloud (Fig. 1). Since all flight tracks are similar, only one track (Oct. 18) is shown in Fig. 1. As cloud and aerosol probe measurements failed during the flight on 5 November and drizzle processes occurred on the flights on 1 November and 2 November, only the observations from other 16 non-drizzling flights are included in this paper.

The aerosol data was obtained by Passive Cavity Aerosol Spectrometer Probe (PCASP-100), which counted and sized particles from 0.1–2.0 µm dry diameter with 20 bins (Zheng et al., 2011; Cai et al., 2013; Twohy et al., 2013). The CCN number concentration was observed by the CCN Spectrometer at a supersaturation of 0.2 % and 0.5% respectively. The cloud data include cloud droplet number concentration ($N_d$, size range: 2.07–40.2 µm with 20 bins) from the Cloud, Aerosol and Precipitation probe (CAS), effective radius of cloud droplets ($R_e$), and liquid water content ($LWC$) from the PVM-100 probe(Gerber et al.,1994). All data sets used in this study are at a frequency of 1 Hz. The calibrations of the onboard instruments were carried out so as to provide standard meteorological variables, aerosol, and cloud observations. Zheng et al. (2011) pointed out that uncertainties of aerosols and cloud measured by these probes are within 15 %. More detailed information about the observation instruments on board the CIRPAS Twin Otter aircraft during VOCALS-REx can be found in Zheng et al. (2010) and Wood et al. (2011).

### 2.2 Data processing

In this study, the data collected near the land, during both take-off and landing, are removed to ensure only the measurements close to Point Alpha (20° S, 72° W) are analysed. The occurrence of clouds is defined by the following criterion, i.e., $LWC > 0.05$ g m$^{-3}$ and $N_d > 15$ cm$^{-3}$. We averaged the CCN number concentrations during the legs within 200 m above the cloud top to obtain the average above-cloud CCN, and within 200 m below the cloud base to obtain the mean sub-cloud





CCN. During the study period, the CCN Spectrometer constantly measured CCN at a supersaturation of 0.2 % except on the

first four flights at a supersaturation of 0.5 %. In order to have a consistent comparison between all flights, we adopted the

method by Zheng et al. (2011) to adjust the CCN concentration from supersaturation of 0.5 % to 0.2 % on the first four flights.

Since the effective diameter of aerosol particle is not measured directly, so we calculated it according to the measurements of

aerosol size distribution based on following equation:

$\mathrm{Da} = \sum n_i d_i^3 / \sum n_i d_i^2$                      (1)

where $n_i$ is the aerosol number concentration in the $i$th bin of PCASP, and $d_i$ represents the arithmetic mean diameter of $i$th

bin.

      To investigate the impact of the entrainment process on cloud properties and aerosol-cloud interaction, we defined

entrainment zone and non-entrainment zone, respectively. Gerber et al. (2005) showed that, in the marine stratocumulus,

entrainment occurs when $LWC$ begins to decrease from the bottom of the cloud. In this manuscript, entrainment and non-

entrainment zone are thus defined as the regions within 20 m above and below the height of maximal $LWC$, respectively. Given

that the two zones are both thin layers, there is little difference in the dynamical and thermos-dynamical conditions. It is

therefore assumed that the difference of cloud microphysical characteristics between the two zones is only caused by

entrainment.

**3 Results**

**3.1 Vertical profiles of aerosol, cloud and meteorological variables**

      The vertical profiles of aerosol, cloud and meteorological variables during 16 flights are scaled by the inversion height

($z_i$) (Fig. 2), which is defined as the height where the vertical gradient of liquid water potential temperature ($\theta_L$) is the largest

(Zheng et al, 2011). $\theta_L$ is conservative for water phase changes, but same as potential temperature when no liquid water exist

(Betts, 1973). This normalization could exclude the variation of $z_i$ between flights, and hence better for exploring the average

BL structure during VOCALS-REx.

      As shown in Fig. 2a, temperature ($T$) decreases sharply with the height within the BL, which is close to dry adiabatic

lapse rate. A strong inversion occurs at the top of the BL, with the average temperature change about 10℃. Due to reduced $T$

and nearly constant water vapor mixing ratio within strong mixing BL, relative humidity ($RH$) increases rapidly with the height

(Fig. 2b). $T$ and $RH$ reach the minimum and maximum, respectively, when $z/z_i$ is close to 0.9. Near the top of the BL (0.9 <

$z/z_i$ < 1.0), the entrainment of the dry and warm air from the free atmosphere aloft results in a slight increase in $T$ and a slight

decrease in $RH$. When $z/z_i > 1$, $T$ increases to about 18 ℃ and $RH$ decreased to about 16 % rapidly (Fig. 2a, b). The vertical

profiles of $T$ and $RH$ are overall consistent with the observations of other marine stratocumulus clouds (Martinet et al., 1994;

Keil and Haywood, 2003). Corresponding to vertical variation of $RH$, the $N_d$ gradually increases with the height, reaches the



maximum when $RH$ is maximum ($z/z_i = 0.9$), and then decreases when $0.9 < z/z_i < 1.0$, indicating that more cloud droplets are

nucleated in high supersaturation. The profile of $LWC$ and $R_e$ is similar to that of $N_d$ (Fig. 2d, e). Fig. 2f reveals that the effective

diameter of aerosol particles ($D_a$) below cloud is larger than that above cloud, which is probably attributed to the different

chemical composition and sources of aerosols. The profile of $CCN/CN$ is similar to that of $D_a$ (Fig. 2g), suggesting that aerosols

with large size are more likely to become CCN (Dusek et al., 2006; Zhang et al., 2011). Larger $D_a$ and $CCN/CN$ are also found

in polluted case than clean cases.

### 3.2 Relationships between aerosol and cloud properties

Aerosol indirect effect is one of the largest uncertainties in current climate assessments. Most studies based on satellite

data employed aerosol optical depth or aerosol index as agents of CCN number concentration to investigate the aerosol-cloud

interactions (Koren et al., 2005, 2010; Su et al., 2010; Tang et al., 2014; Ma et al., 2014, 2018; Wang et al., 2014, 2015;

Saponaro et al., 2017). However, not all aerosols on the vertical column are actually involved in cloud formation, thus this

assumption is relatively rough. Several studies revealed that aerosols have little effect on cloud properties when aerosol and

cloud layers are clearly separated (Costantino and Bréon, 2010, 2013; Liu et al., 2017). In this study, the impact of CCN

number concentration near cloud layer, e.g. below and above cloud respectively, on cloud properties is assessed.

The relationships between sub-cloud CCN number concentration (*sub-CCN*) and cloud properties during all flights are

shown in Fig. 3. The red dots signify the ten flights with typical well mixed boundary layer and non-drizzling cases, which

have relatively similar meteorological conditions, such as similar inversion heights, and the jump of potential temperature and

total water mixing ratio across the inversion (Zheng et al., 2010), and thus can be used to isolate the response of cloud properties

to aerosol perturbations. The blue dots represent the other cases, in which the conditions except typical well mixed boundary

layer and non-drizzling, such as strong wind shear within the BL, moist layers above clouds, strong decoupled BL and so on,

are involved (Table 2). In the case of typical well mixed boundary with non-drizzling, both $LWC$ (Fig. 3a) and $N_d$ (Fig. 3b)

exhibit the positive relationships with *sub-CCN*, with correlation coefficients of 0.60 and 0.79, respectively, while $R_e$ has no

evident correlation with *sub-CCN* (Fig. 3c). This may imply that the increase of $LWC$ induced by *sub-CCN* is mainly caused

by increasing $N_d$ instead of $R_e$. Fig. 3d indicates a positive correlation between cloud depth and *sub-CCN*, with correlation

coefficient of 0.71. As cloud top height is mainly determined by the temperature inversion condition, there is no obvious

correlation between cloud top height and sub-CCN, with correlation coefficient of only –0.13 (Fig. 3e). However, the

correlation coefficient between cloud base height and *sub-CCN* is –0.69 (Fig. 3f), suggesting that CCN thickening cloud is

mainly induced by lowering cloud base. It is noted that the above conclusions are only valid in the typical mixed boundary

layer. In other cases (i.e. blue dots), the impacts of aerosols on the cloud is not evident due to large difference in the

meteorological conditions and the boundary layer structure.





Compared to sub-cloud CCN, the influence of above-cloud CCN on cloud properties is very weak. The absolute values

of correlation coefficient between above-cloud CCN number concentration (*abv-CCN*) and cloud properties are all less than

0.4 (figure omitted), none of which pass the significance test ($\alpha$ = 0.05). In this study, above-cloud aerosol number

concentration is very low ($129.8 \pm 60.1$ cm$^{-3}$) and the inversion capped the cloud top is extremely strong, which weakens the

mixing of the aerosol with cloud layer and hence the effects of aerosol on cloud properties. Some previous studies based on

aircraft observation for stratocumulus clouds also found that $N_d$ exhibits a significantly positive correlation with *sub-CCN*, but

no correlation with *abv-CCN* (Martin et al., 1994; Hudson et al., 2010; Hegg et al., 2012).

     In order to investigate cloud formation in different aerosol loadings, the most polluted (Oct. 19) and the cleanest (Nov.

09) cases with aerosol concentrations of $647.78 \pm 60.47$ cm$^{-3}$ and $268.97 \pm 35.67$ cm$^{-3}$, respectively, are selected in this study.

Vertical profiles for the two cases are highlighted in Fig. 2, showing that $N_d$ and *LWC* in polluted case are larger than those in

clean one, but $R_e$ remains the same. The low aerosol concentrations under the clean case inhibit the increase of $N_d$ with *LWC*

(Fig. 4a), which hence promotes the rapid increase of $R_e$ with *LWC* (Fig. 4b). On the contrary, there are enough particles which

may potentially activated into cloud droplets under the polluted case, thus $N_d$ increases rapidly with *LWC*. As the certain

amount water is shared by large amount particles, the increase of $R_e$ is limited. It is suggested that the increase of *LWC* is

mainly contributed by $N_d$ instead of $R_e$ when aerosol concentrations is high, in which large number of cloud droplets are formed

with smaller size, but the opposite is true when aerosol concentrations is low. The result is consistent with the study in Beijing

by Zhang et al. (2011), but the difference of cloud formation between clean and polluted conditions is less evident, which is

probably attributed to the much lower aerosol concentration difference between clean and polluted cases in this study (about

400 cm$^{-3}$) than that in Zhang et al. (2011) (about 7000 cm$^{-3}$).

### 3.3 Cloud droplet formation and its controlling factors

Sub-cloud CCN is considered as a good proxy for the aerosol entering cloud. However, during actual flight, it is difficult

to collect enough samples of sub-cloud CCN and cloud droplets simultaneously, which may result in uncertainty in statistical

analysis. This limitation can be overcome by employing interstitial aerosols. Interstitial aerosols are particles observed in-

cloud that either never activate into cloud droplets or have been activated but then return into aerosols after evaporation of

cloud droplet. Kleinman et al. (2012) pointed out that the number concentration of interstitial aerosol ($N_i$) can be obtained

either directly from the observation of in-cloud aerosols, or indirectly from a number balance between sub-cloud and in-cloud

particles. In this study, the interstitial aerosol properties are derived from direct measurements in cloud. By employing aircraft

observations over both land and ocean, Gultepe et al. (1996) found that the difference of the number concentration between

total in-cloud particles ($N_d + N_i$) measured directly and sub-cloud aerosols is very small. It is thus assumed that total in-cloud

particles can characterize the overall level of in-cloud aerosol concentration before activation. The flight on Oct. 18 is singled

out as a case study to support this assumption (Fig. 5). It is shown that the number concentrations of sub-cloud aerosols and





total in-cloud particles are very close, with the values of $583.7 \pm 55.4$ cm$^{-3}$ and $567.4 \pm 59.1$ cm$^{-3}$ respectively. Similar results are also found in other flights. The average ratio of $N_d + N_i$ to sub-cloud aerosol concentration during all flights is 0.94, which is much smaller than the value (1.29) found by Kleinman et al. (2012) based on G-1 aircraft during VOCALS-REx. Therefore, the observation of interstitial aerosols in this study is unlikely to be significantly interfered by factors such as cloud droplet

shatter and cloud droplet evaporation due to instrument heating, as discussed by Kleinman et al. (2012), which has the potential to create more extra aerosols in-cloud.

The relations between $N_d$ and $N_d + N_i$ during 16 non-drizzling flights are shown in Fig. 6, in which the colors represent in-cloud vertical velocities. Positive correlations between $N_d$ and $N_d + N_i$ are found in all flights, representing the aerosol-cloud interaction (IPCC, 2001, 2007, 2013; Hegg et al., 2012). In addition, the effect of dynamical conditions on cloud droplet

formation is evident. As presented in Fig. 6, data are close to the 1:1 line when vertical velocity is relatively large, namely in-cloud aerosols are almost entirely activated into cloud droplets. However, data deviate from the 1:1 line when vertical velocity is small or negative. For example, the ratio of $N_d$ to $N_d + N_i$ with vertical velocity greater than 1 m s$^{-1}$ is $0.84 \pm 0.12$, which is much larger than that with vertical velocity less than $-1$ m s$^{-1}$ ($0.64 \pm 0.14$). This is possibly attributed to high supersaturation caused by the adiabatic uplift under conditions with large vertical velocity. High supersaturation not only induces more aerosols

to reach critical supersaturation and then activate into cloud droplets, but also inhibits cloud droplet evaporation.

In addition to dynamical conditions, aerosol microphysical properties, such as size distribution and chemical components, also affect activation process significantly (Nenes et al., 2002; Lance et al., 2004; Ervens et al., 2005;Dusek et al., 2006; McFiggans et al., 2006; Zhang et al.,2011; Almeida et al., 2014; Leck and Svensson, 2015). Since part of aerosols in the cloud have activated to cloud droplets, it is difficult to obtain the information of aerosol size before activation. According to Köhler

theory, the critical supersaturation of aerosol with large size is relatively low, and thus activate preferentially, i.e. the effective diameter of interstitial aerosol ($D_i$) is smaller than that of initial aerosols before activation. Li et al. (2011) compared the difference of size distribution between interstitial aerosol and aerosols that have been activated to cloud droplets, and found that the peak diameter of the former (0.45 μm) was much smaller than that of the latter (0.8 μm). It can be thus inferred that the size of aerosols activated to cloud droplets, and thus the size of initial aerosols would be larger with the increase of $D_i$,

though the quantitative relation depends on in-cloud dynamics. Therefore, it is assumed that, when compared with the data measured at different sampling locations during flight, the size of interstitial aerosol can still represent the size of initial aerosols before activation to some extent. As indicated in Fig. 7, the larger $D_i$ is, the closer the data is to the 1:1 line, i.e. the higher proportion of cloud droplets in total in-cloud particles ($N_d /(N_d + N_i)$) is. The averaged $N_d /(N_d + N_i)$ is $0.76 \pm 0.13$ when $D_i$ is larger than 1.0 μm, but only $0.64 \pm 0.23$ when $D_i$ is less than 0.5 μm. It is because that aerosols with large size are more likely to be activated into cloud droplets. Additionally, as larger aerosol particles form to larger cloud droplets (Twohy et al.,





1989, 2013) that are relatively difficult to evaporate, large particles can also inhibit cloud droplet evaporation to a certain extent.

### 3.4 Dispersion effect

In addition to modulating the cloud droplet number concentration, aerosols also affect the shape of cloud droplet size spectrum (referred to as "dispersion effect") and thereby cloud albedo (Liu and Daum, 2002). When the dispersion effect is taken into account, the estimated aerosol indirect forcing could be either reduced (Liu and Daum, 2002; Peng and Lohmann, 2003; Kumar et al., 2016; Pandithurai et al., 2012) or enhanced (Ma et al., 2010), i.e., dispersion effect could act to either offset or enhance the well-known Twomey effect, which mainly depends on the sensitivity of the relative dispersion ($\varepsilon$, the ratio of the standard deviation to the mean radius of the cloud droplet size distribution) on aerosol number concentration ($N_a$).

However, the relationship between $\varepsilon$ and $N_a$ still remains large uncertainty. Table 1 shows that the observed correlations between $\varepsilon$ and $N_d$ (or $N_a$) can be positive, negative, or not evident. Different relations are indicative of the fact that the effect of aerosol on $\varepsilon$ is often intertwined with effects of other factors, especially cloud dynamical conditions (Pawlowska et al., 2006; Lu et al., 2012). In this section, the relationship between $\varepsilon$ and $N_d$ based on the in-flight and the flight-averaged data are discussed respectively in order to distinguish the influences of aerosol and cloud dynamics on $\varepsilon$.

Within an individual flight, aerosol number concentration and chemical components can be assumed to be similar, providing an opportunity to focus on the effect of cloud dynamics to the extent possible. Here, we employ vertical velocity ($w$, m s$^{-1}$) as a proxy for cloud dynamical condition. As shown in Fig. 8, the correlations between $\varepsilon$ and $N_d$ based on in-flight data is significantly negative during all 16 non-drizzling flights, which is mainly modulated by $w$, i.e., larger $w$ corresponds to a smaller $\varepsilon$ but larger $N_d$. High supersaturation leads to more cloud droplets to activate and grow to the same size (i.e., narrow the droplet spectrum) when $w$ is relative large, but a portion of cloud droplets may evaporate into smaller size and even deactivate into interstitial aerosols when $w$ is small or even negative, resulting in the decrease of $N_d$ and the broadening of the droplet spectrum.

It is interesting to see from Table 1 that the correlations between $\varepsilon$ and $N_d$ based on in-flight data are generally negative, while the one based on the flight-averaged data could be either positive, negative, or even uncorrelated. The uncertain relationships of the later may result from variations of the strength of cloud dynamic between flights, which would disrupt or even cancel the real influence of aerosol on relative dispersion (Peng et al., 2007; Lu et al., 2012). However, many previous studies did not take the difference of cloud dynamics in flights into account when correlating $\varepsilon$ and $N_d$, which could result in some degree of overestimation or underestimation of dispersion effect. In this study, data in all flights were sampled over the same location, i.e., Point Alpha, which can reduce the difference of dynamic conditions caused by variations of horizontal sampling location. In addition, we also distinguish the flights of typical mixed boundary layer and the others to ensure relatively similar meteorological conditions (see section 3.2). Fig. 9 shows the probability distribution function of $w$ with mean values



and standard deviations for 16 non-drizzling flights. It can be found that, except for other cases (gray shadow; especially Oct. 24, Oct. 29, Nov. 8, and Nov. 13), the difference of in-cloud dynamics between typical well mixed boundary flights is very small, which confirms the assumption of similar meteorological conditions. As indicated in Fig. 10a, $\varepsilon$ and $N_d$ are positively

correlated (correlation coefficient of 0.29 and the slope of $1.9 \times 10^{-4}$) in the case of the typical well mixed boundary, indicating that aerosol increases $\varepsilon$ and $N_d$ at the same time. However, correlation coefficient and slope reduce to 0.11 and $7.7 \times 10^{-5}$, respectively in the all cases (i.e., not to constrain w), implying that the influence of aerosol on $\varepsilon$-$N_d$ relationship tends to be weaker after intertwined with effects of cloud dynamics. Although the perturbations of cloud dynamics have been eliminated as far as possible, $N_d$ is still likely determined by both aerosols number concentrations and updraft velocity together. Therefore,

similar statistical analysis are also conducted for sub-cloud CCN. The relationship between $\varepsilon$ and sub-cloud CCN is similar to that between $\varepsilon$ and $N_d$, but, as expected, the correlation coefficient (slope) in the case of typical well mixed boundary and all cases increase to 0.67 ($3.1 \times 10^{-4}$) and 0.31 ($2.1 \times 10^{-4}$), respectively (Fig. 10b).

### 3.5 Entrainment in stratocumulus

Entrainment is a key process in the clouds, which plays an important role in the formation and evolution of clouds and

the change of droplet spectrum, as well as aerosol indirect effect (Chen et al., 2014, 2015; Andersen and Cermak, 2015). The nature of entrainment is related to the cloud type. Entrainment in cumulus is primarily lateral with strong dilution of the cloud, which induces $LWC$ to decrease rapidly to about 20% of its adiabatic value (Warner, 1955). Entrainment in stratocumulus proceeds from the top and affects mostly a thin layer (Gerber et al., 2005), whose dilution effect is much weaker than that in cumulus (Warner, 1955, 1969a, 1969b; Blyth et al., 1988; Gerber et al., 2008; Burnet and Brenguier, 2007; Haman et al., 2007).

Aircraft observations of marine stratocumulus showed that the vertical profile of $LWC$ is essentially same as the adiabatic profile, i.e. the cloud is almost adiabatic (Keil and Haywood, 2003).

In order to explore the entrainment in stratocumulus during VOCALS-REx, we firstly compared the differences of cloud microphysics between entrainment and non-entrainment zone near the cloud top. Here, entrainment and non-entrainment zone are defined as the regions within 20 m above and below the height of maximal $LWC$, respectively. As anticipated, adiabatic

fraction ($AF$, the ratio of the measured $LWC$ to its adiabatic value) in entrainment zone ($AF_{ent}$) is generally lower than that in non-entrainment zone ($AF_{non\text{-}ent}$), with the mean values of all flights of 0.64 and 0.77 respectively (Table 2), which further confirms the rationality in dividing the two zones. Compared with non-entrainment zone, the peak diameters of cloud droplets in entrainment zone has little change (Fig. 11), and the effective diameters of cloud droplet ($D_e$) increases only by 1.8 % (Table 2). However, $N_d$ and $LWC$ decrease significantly by 28.9 % and 24.8% respectively on average (Table 2), especially during

flights on Oct. 18, Nov. 04, Nov. 09 and Nov. 13, $N_d$ decreases by 60.1 %, 56.3 %, 56.1 % and 59.2 %, and $LWC$ decreases by 55.7 %, 62.1 %, 55.8 % and 58.7 %, respectively (Table 2). It is suggested that dry and warm air entrained from cloud top dilutes $N_d$ and $LWC$ by a similar amount, while the size of droplets is relatively unaffected, which is thought as extreme



inhomogeneous entrainment-mixing process. Although it is still unclear whether the entrainment-mixing mechanism is predominantly homogeneous, inhomogeneous, or in between (Andrejczuk et al., 2009; Lehmann et al., 2009), previous studies

showed that stratocumulus is, in general, dominated by the inhomogeneous (Pawlowska et al., 2000; Burnet and Brenguier, 2007; Haman et al., 2007; Lu et al., 2011; Yum et al., 2015).

As shown in previous studies, nucleation of cloud droplet mainly occurs near cloud base, and sub-cloud aerosols are the major source of cloud droplets (Pinsky and Khain, 2002; Ghan et al., 2011). However, de Rooy et al., (2013) pointed out that entrainment mixing at the cloud edge and cloud-top contribute significantly to the amount of entrained air and hence aerosols.

Therefore, activation of aerosols is not restricted to the cloud base, where the central updraft enters the cloud (primary activation). Slawinska et al. (2012) found that a significant part (40%) of aerosols is activated above cloud base (secondary activation), which is dominated by entrained aerosols. By using large-eddy simulations (LES), Hoffmann et al. (2015) suggested that, in a shallow cumulus, sub-cloud aerosols and laterally entrained aerosols contribute to all activated aerosols inside the cloud by fractions of 70% and 30%, respectively. Although entrainment in stratocumulus, discussed in this

manuscript, is weaker than that in cumulus, entrained aerosols is still a possible source of cloud droplets. In this study, the flight on Oct. 18 with strong entrainment is chosen to investigate the difference of cloud droplet formation between entrainment and non-entrainment zone. As presented in Fig. 12a, the probability in entrainment zone is significantly higher than that in non-entrainment zone when $D_i < 0.75$ μm, but the opposite is true when $D_i > 1.1$ μm. This result indicates that small particles are entrained into cloud from the top (Fig. 2f) and large particles are detrained out cloud at the same time. The decrease of $D_i$

by 0.18 μm may inhibit aerosol activation into cloud droplet. Furthermore, dry and warm air entrained from the top reduces the relative humidity by 8.8 % on average (Fig. 12b), and accelerates the cloud droplets evaporation. As a result, $N_d /(N_d + N_i)$ in entrainment zone ($0.56 \pm 0.22$) is much lower than that in non-entrainment zone ($0.73 \pm 0.13$) (Fig. 12c). It is also noted that the relative dispersion in entrainment zone is overall larger than that in non-entrainment zone (Fig. 12d), implying that smaller aerosol particles and drier air entrained from the top could broaden cloud droplet spectrum by influencing nucleation

and evaporation of cloud droplets. Some previous observations also showed that $\varepsilon$ with low $AF$ tends to be larger than that with high $AF$, and attributed it to the effect of entrainment mixing (Pawlowska et al., 2006; Lu et al., 2009). According to the discussion in Sect. 3.3, although the impact of above-cloud aerosol on whole cloud is much weaker than sub-cloud aerosols, the entrainment of above-cloud aerosols may affect the cloud droplets nucleation, and hence change cloud properties near the cloud top to some extent.

**4 Summary**

By using in situ aircraft data collected by CIRPAS Twin Otter aircraft at Point Alpha during VOCALS-REx from 16 October to 13 November 2008, we investigated the interaction between aerosol and marine stratocumulus over the southeast



Pacific Ocean, especially the dispersion effect. We also explored the entrainment process near the top of stratocumulus and its impacts on cloud properties and aerosol-cloud interaction.

Vertical profiles of aerosol, cloud and meteorological variables presented that the BL is well mixed and capped by a sharp inversion during 16 non-drizzling flights. Cloud variables, such as $LWC$, $N_d$, and cloud depth, are all positive correlated with sub-cloud CCN number concentration, having the correlation coefficients of 0.60, 0.79 and 0.71, respectively. No evident correlation was found between cloud properties with above-cloud CCN number concentrations. This is mainly due to low aerosol number concentrations above-cloud ($129.8 \pm 60.1$ cm$^{-3}$) and the extremely strong inversion capped the cloud top, which

inhibits the mixing of the above-cloud aerosol with cloud layer. Therefore, the influence of above-cloud CCN on cloud properties is very weak compared to sub-cloud CCN. Additionally, the comparison of cloud formation under different aerosol number concentrations conditions suggested that the increase of $LWC$ is probably contributed by $N_d$ instead of $R_e$ in the polluted case due to abundant CCN, in which more but smaller cloud droplets form, while the opposite is true in the clean case.

The results showed that both dynamical condition and aerosol microphysical properties have significant effects on cloud

droplet formation. In the case of large vertical velocity and aerosol size, the proportion of cloud droplet of total in-cloud particles is relatively high (e.g. $0.84 \pm 0.12$ and $0.76 \pm 0.13$, respectively), i.e., cloud droplets are easier to form. Although chemical components of aerosol is also critical to cloud droplet formation (Nenes et al., 2002; Lance et al., 2004; Ervens et al., 2005; McFiggans et al., 2006; Wang et al., 2008; Almeida et al., 2014), this was not discussed in this study due to unavailable measurements.

The correlations between $\varepsilon$ and $N_d$ based on the in-flight data, used to represent $w$-induced correlation, is significantly negative, while the correlations derived from flight-averaged data (i.e., aerosol-induced correlation) is positive. This implies that an increase in aerosol concentration tends to increase $\varepsilon$ and $N_d$ at the same time, while an increase in $w$ often increases $N_d$ but decreases $\varepsilon$, which is in agreement with theoretical analysis (Liu et al., 2006). After constraining the differences of cloud dynamics between flights, positive correlation between $\varepsilon$ and $N_d$ become stronger, indicating that perturbations of $w$ could

weaken the influence of aerosol on $\varepsilon$, and hence may result in an underestimation of aerosol dispersion effect. Thus, it requires more attention to isolate the response of relative dispersion to aerosol perturbations from dynamical effects when investigating aerosol dispersion effect and estimating aerosol indirect forcing.

The entrainment in stratocumulus is overall quite weak, and close to adiabatic in some case. In this study, the difference of cloud microphysics between entrainment and non-entrainment zone indicated that the entrainment in stratocumulus is

mostly dominated by extreme inhomogeneous entrainment-mixing mechanism. On average, the entrainment reduced $N_d$ and $LWC$ by 28.9 % and 24.8 %, respectively, while had little effect on $D_e$ (only increases by 1.8 %). During flights on Oct. 18, Nov. 04, Nov. 09 and Nov. 13, the entrainment is relatively strong and dilutes $N_d$ and $LWC$ by about 50 %. In entrainment zone, the smaller aerosols and drier air entrained from the top result in the smaller $N_d /(N_d + N_i)$ ($0.56 \pm 0.22$) than that in non-





entrainment zone (0.73 ± 0.13). This implies that entrainment may significantly influence cloud droplet formation and hence

cloud properties near the top by both inhibiting aerosol activation and promoting cloud droplets evaporation. Furthermore, we

also found that the relative dispersion in entrainment zone is larger than that in non-entrainment zone. As stated above, although

entrainment in stratocumulus is much weaker than that in other cloud types, e.g., cumulus (Warner, 1955, 1969a, 1969b; Blyth

et al., 1988; Gerber et al., 2008; Burnet and Brenguier, 2007; Haman et al., 2007), entrainment in stratocumulus still impact

cloud droplet formation near cloud-top significantly by entraining ambient dry air as well aerosols with physical and chemical

properties different from that in cloud. Therefore, entrainment is important to take into account in studying aerosol-cloud

interaction, even in stratocumulus with relatively weak entrainment. However, a quantitative contribution of entrained dry air

and aerosols to cloud droplet formation, is difficult to determine only using pure aircraft measurements.

*Data availability.* The aircraft measurements data during VOCALS-REx was obtained from the public ftp at

http://data.eol.ucar.edu/master_list/?project=VOCALS.

*Competing interests.* The authors declare that they have no conflict of interest.

*Acknowledgements.* We are grateful for the dedicated efforts of several support staff and scientists in making the observations
from the CIRPAS Twin Otter during VOCALS-REx. We also thank Prof. Bruce Albrecht in University of Miami for kindly
providing the aerosol, cloud and meteorological variables observations, which are the basis of this manuscript. This study is
supported by the National Natural Science Foundation of China grants (41475005 and 41675004).

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



**Table 1. Correlations between $\varepsilon$ and $N_d$ ($N_a$) from observation studies.**

| Observations | Observation type | Location | Data for correlation analysis | Correlation |
|---|---|---|---|---|
| Liu and Daum, 2002 | Aircraft | Ocean & coast | Flight-averaged | Positive |
| Peng and Lohmann, 2003 | Aircraft | Coast | Flight-averaged | Positive |
| Pawlowska et al., 2006 | Aircraft | Ocean | In-flight<br>Flight-averaged | Negative<br>Positive |
| Zhao et al., 2006 | Aircraft | Land, ocean, and coast | In-flight | $\varepsilon$ converges to a small range of values with increasing $N_d$ |
| Lu et al., 2007 | Aircraft | Ocean | In-flight<br>Flight-averaged | Negative<br>None for $N_d$; Positive for $N_a$ |
| Lu et al., 2012 | Aircraft | Land | In-flight<br>Flight-averaged | Negative<br>Negative |
| Hudson et al., 2012 | Aircraft | Ocean | Flight-averaged | Negative |
| Ma et al., 2012 | Aircraft | Land | Flight-averaged | Negative |
| Pandithurai et al., 2012 | Aircraft | Land | Flight-averaged | Positive |
| Kumar et al., 2016 | ground-based | Land | — | Positive |






**Table 2. Flight information and parameters that represent the properties of entrainment during all 16 non-drizzling flights.**

| Flight number | RF01 | RF02 | RF03 | RF04 | RF05 | RF06 | RF07 | RF08 | RF09 |
|---|---|---|---|---|---|---|---|---|---|
| Date | 10.16 | 10.18 | 10.19 | 10.21 | 10.22 | 10.24 | 10.26 | 10.27 | 10.29 |
| BL type | **Typical** | **Typical** | **Typical** | **Typical** | **Typical** | **Other** Wind shear | **Typical** | **Typical** | **Other** Decoupled |
| $P_{LWC}$[a] | 25.8 | 55.7 | 33.4 | 24.8 | 24.6 | 29.3 | -2.7 | 11.2 | 3.1 |
| $P_{Nd}$[b] | 32.1 | 60.1 | 30.1 | 38.6 | 28.2 | 34.4 | 4.9 | 19.6 | 6.4 |
| $P_{De}$[c] | -1.9 | -5.7 | 0.9 | -6.7 | -1.9 | -0.1 | -4.1 | -2.4 | -1.8 |
| $AF_{ent}$[d] | 0.77 | 0.52 | 0.58 | 0.85 | 0.49 | 0.52 | 0.51 | 0.76 | 0.81 |
| $AF_{non-ent}$[e] | 0.95 | 0.84 | 0.82 | 0.77 | 0.74 | 0.78 | 0.73 | 0.82 | 0.80 |

| Flight | RF10 | RF11 | RF12 | RF13 | RF14 | RF15 | RF16 | Total |
|---|---|---|---|---|---|---|---|---|
| Date | 10.30 | 11.04 | 11.08 | 11.09 | 11.10 | 11.12 | 11.13 | |
| BL type | **Other** Wind shear | **Other** Wind shear, Decoupled | **Other** Decoupled | **Typical** | **Typical** | **Typical** | **Other** Moisture above | |
| $P_{LWC}$ | 10.5 | 62.1 | 2.5 | 55.8 | 2.9 | -1.8 | 58.7 | 24.8 |
| $P_{Nd}$ | 7.6 | 56.3 | 24.0 | 56.1 | -1.6 | 7.5 | 59.2 | 28.9 |
| $P_{De}$ | 0.2 | 4.4 | -8.4 | -2.1 | 3.4 | -2.5 | -1.2 | -1.8 |
| $AF_{ent}$ | 0.73 | 0.66 | 0.84 | 0.28 | 0.70 | 0.67 | 0.56 | 0.64 |
| $AF_{non-ent}$ | 0.82 | 0.97 | 0.77 | 0.50 | 0.79 | 0.60 | 0.64 | 0.77 |

[a, b, c] $P_{LWC}$, $P_{Nd}$, and $P_{De}$ are the percentages of reduction in $LWC$, $N_d$ and $D_e$ within entrainment zone relative to non-entrainment zone.(unit: %)

560        [d, e] $AF_{ent}$ and $AF_{non-en}$ are adiabatic fraction in entrainment zone and non-entrainment zone, respectively. Here, adiabatic fraction is defined as the ratio of the measured to its adiabatic $LWC$ that is calculated using pressure and temperature near cloud base.

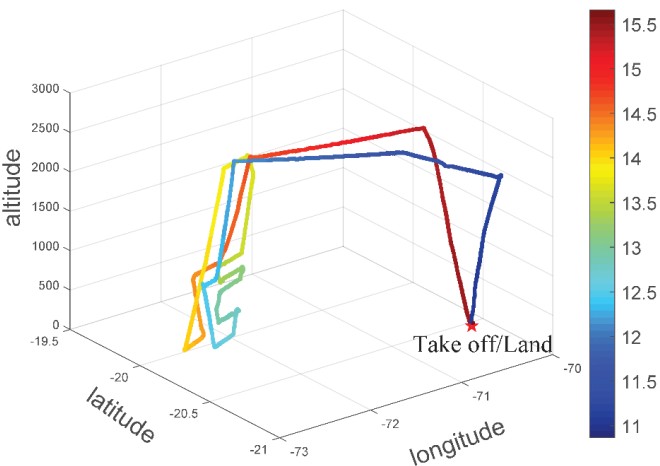

565        **Fig. 1. The flight track in Oct. 18, and the colors represent flight time in hour (UTC).**



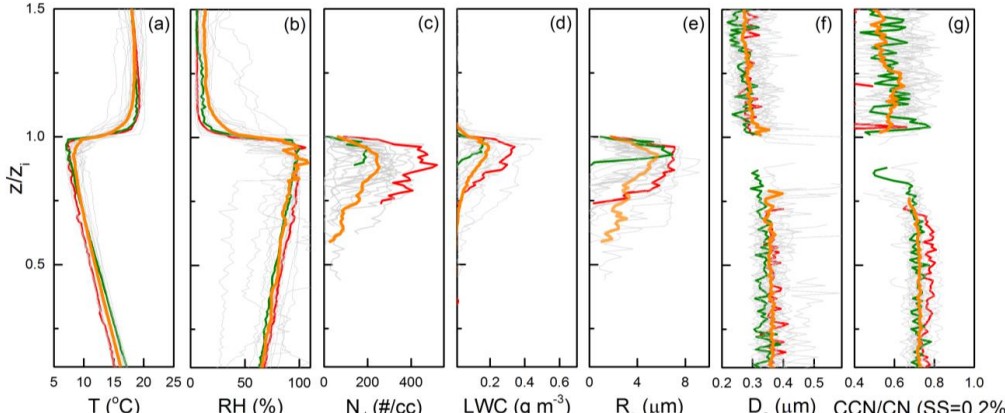

**Fig. 2. Vertical profiles scaled by the inversion height. (a) temperature (K); (b) relative humidity (%); (c) cloud droplet number concentration (cm$^{-3}$); (d) liquid water content (g m$^{-3}$); (e) effective radius of cloud droplets (µm); (f) effective diameter of aerosols (µm), and (g) the number concentration ratio of CCN to aerosols for all 16 non-drizzling flights. The gray lines show all individual flights, and the orange lines indicate the average profiles. The red and green lines represent the polluted (Oct. 18) and clean (Nov. 9) cases, respectively.**

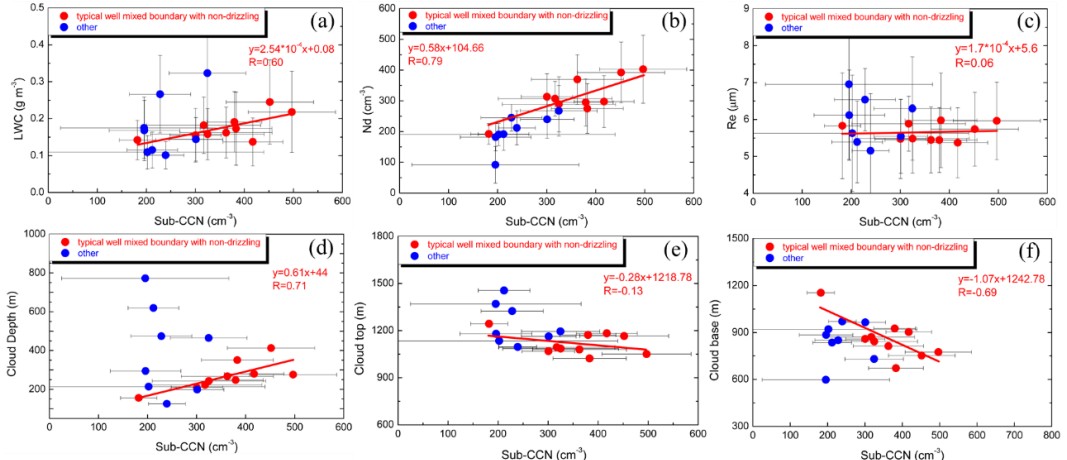

**Fig. 3. (a) LWC (g cm$^{-3}$); (b) $N_d$ (cm$^{-3}$); (c) $R_e$ (µm); (d) cloud depth (m); (e) cloud top height (m); (f) cloud base height (m) as a function of sub-cloud CCN concentrations (SS=0.2%) for all flights. The error bars through these symbols indicate the standard deviation. Red symbols are the typical well mixed boundary with non-drizzling discussed in Zheng et al. (2011), and blue symbols for others.**



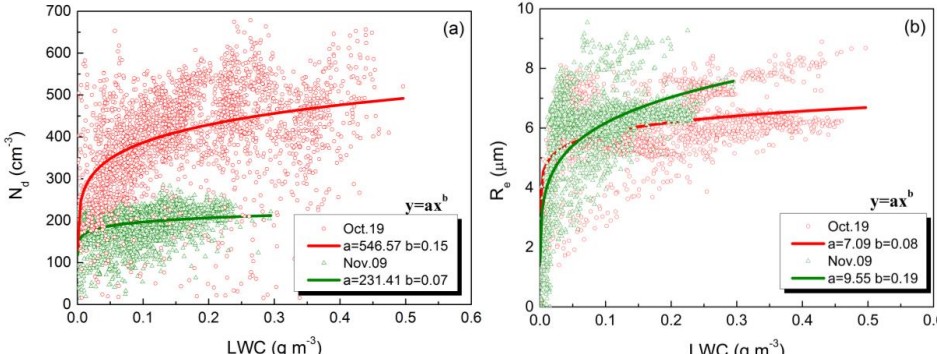

**Fig. 4. Correlations between (a) $N_d$ (cm⁻³), (b) $R_e$ (μm) and $LWC$ (g m⁻³) for clean (green) and polluted (red) cases, respectively.**


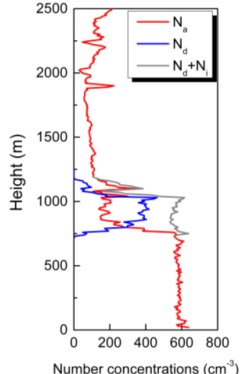

**Fig. 5. Vertical profiles of number concentrations of aerosols ($N_a$), cloud droplets ($N_d$) and total in-cloud particles ($N_d + N_i$) during**

**the flight on Oct. 18.**



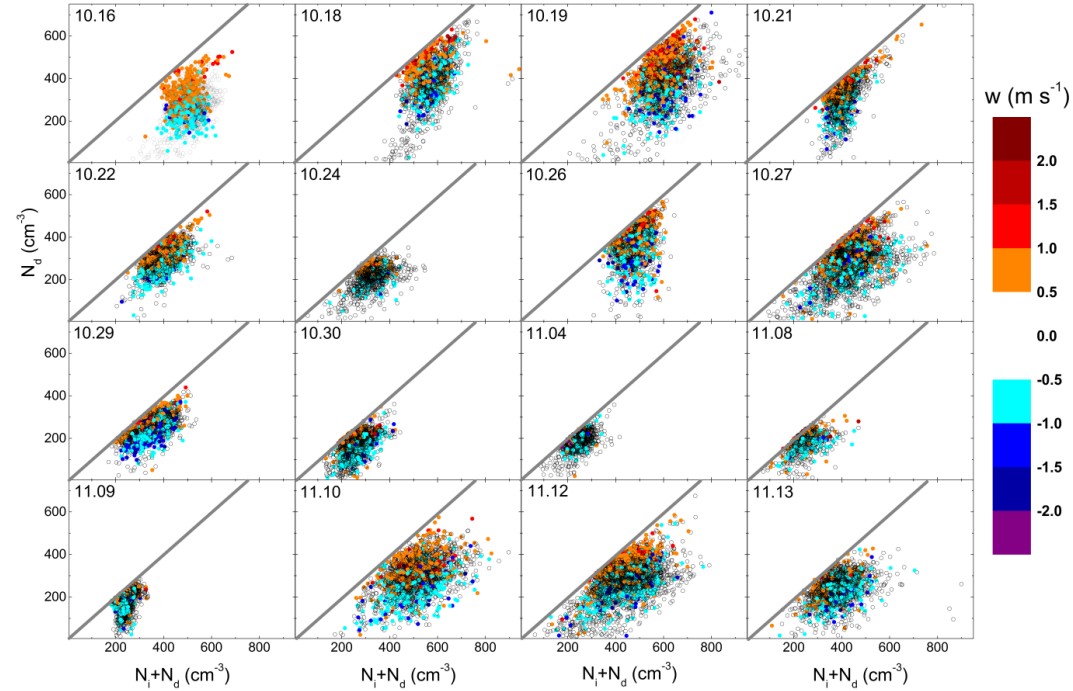


**Fig. 6. Relationships between $N_d$ and $N_i + N_d$ during all 16 non-drizzling flights. The colors represent in-cloud vertical velocities (m s$^{-1}$), and gray line is 1:1 line.**



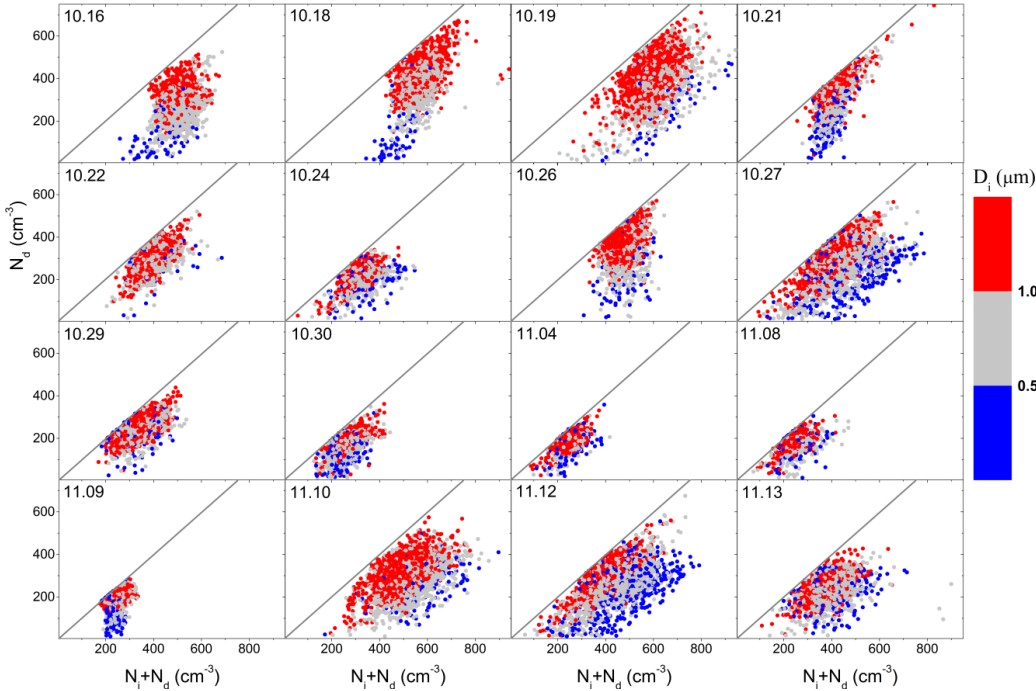

**Fig. 7. Same as Fig. 6, but the colors represent the effective diameter of interstitial aerosol ($D_i$) (μm)**



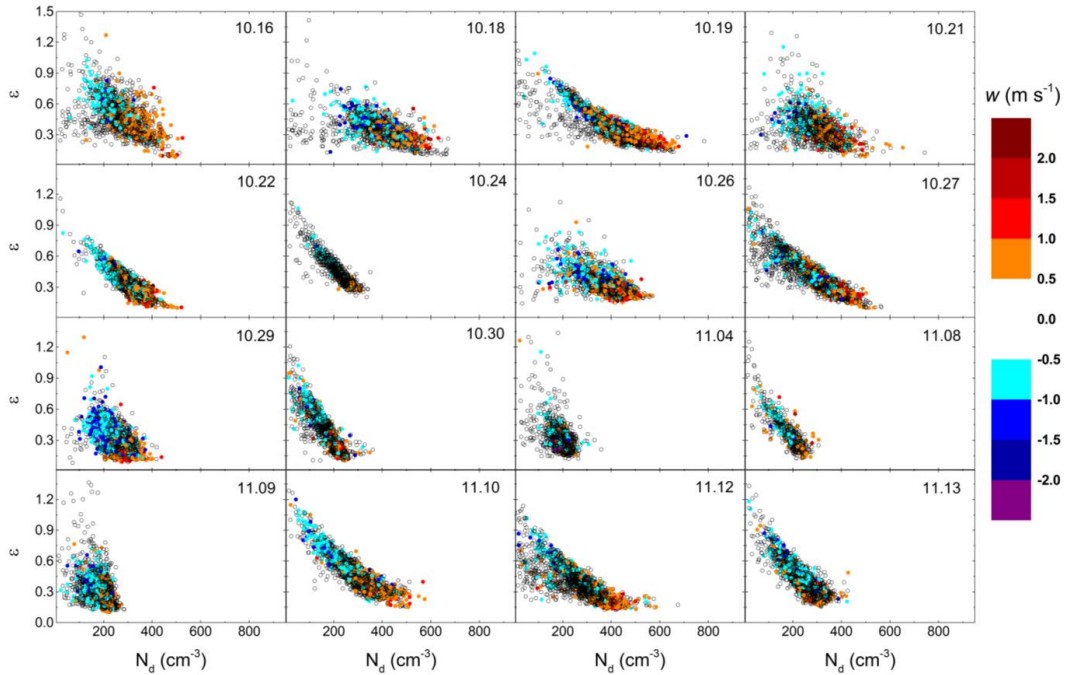

**Fig. 8. Relationships between relative dispersion (ε) and $N_d$ during all 16 non-drizzling flights, in which the colors representing in-**

**cloud vertical velocities (m s⁻¹).**

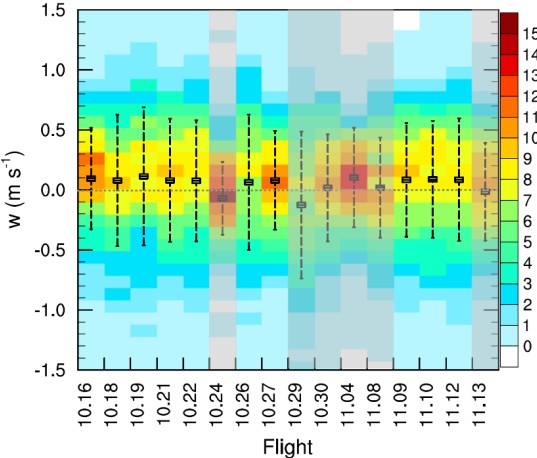

**Fig. 9. Probability distribution function (units: %) of vertical velocity (w) for 16 non-drizzling flights. Black symbols are mean values**

**of w, and error bars through these symbols indicate the standard deviation. Gray shadow represents the flights other than typical**

**well mixed boundary with non-drizzling.**





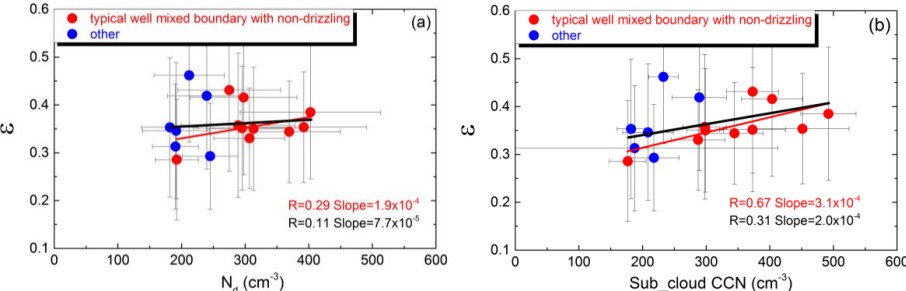

**Fig. 10. Relative dispersion (ε) as a function of (a) $N_d$ and (b) sub-cloud CCN concentrations ($SS$=0.2%) for all flights. The error bars**

**through these symbols indicate the standard deviation. Red symbols are the typical well mixed boundary with non-drizzling, and**

**blue symbols for others. Red (black) texts are the correlation coefficient and slope for typical well mixed cases (all cases).**

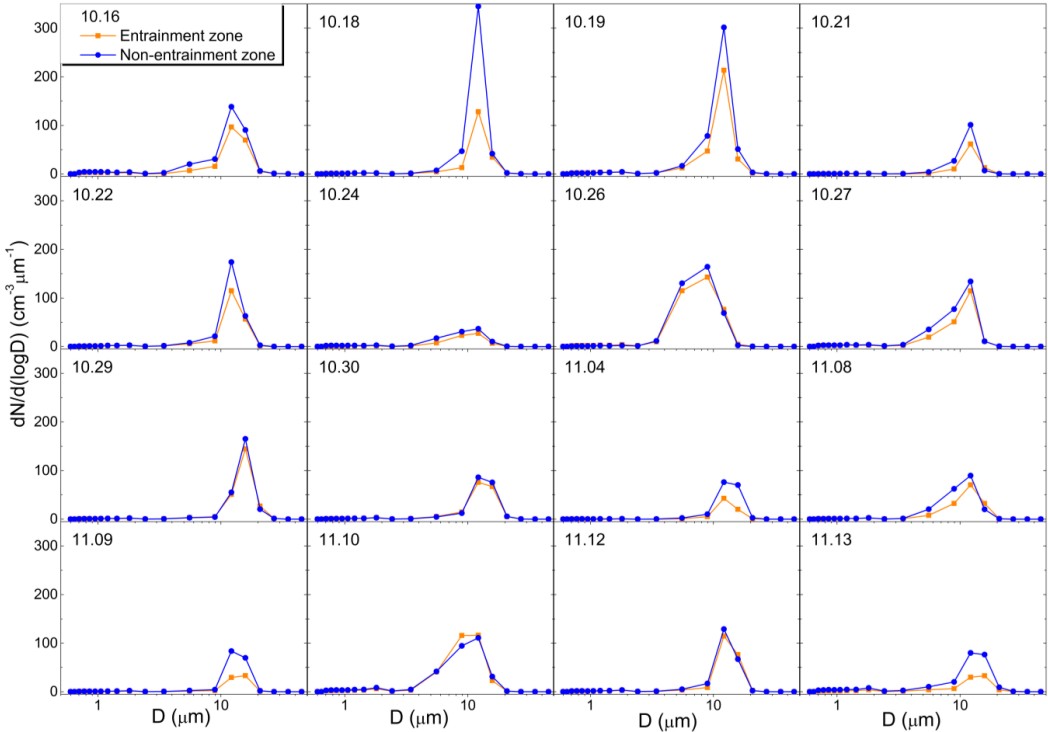

**Fig. 11. Number size distributions of cloud droplets in entrainment (yellow) and non-entrainment zone (blue) during all 16 non-**

**drizzling flights.**



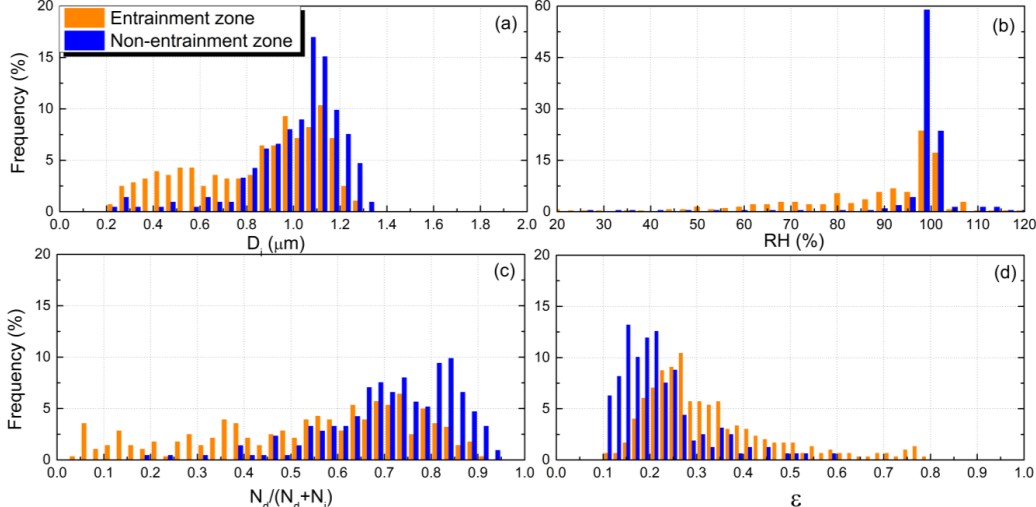

**Fig. 12. Probability density functions of (a) $D_i$ (μm), (b) $RH$ (%), (c) $N_d/(N_d + N_i)$, and (d) $\varepsilon$ in entrainment (yellow) and non-**

**entrainment zone (blue) during the flight on Oct. 18**