# Peer review of "Exploring aerosol cloud interaction using VOCALS-REx aircraft measurements"

_Atmospheric Chemistry and Physics, 2018_

## Referee Comment (RC1) · Anonymous Referee #1 · 22 Nov 2018

article

**1   Overview Comments**

This paper uses data from a field campaign in the south eastern Pacific to investigate the aerosol dispersion effect and entrainment in stratocumulus clouds. The cases have been described in other work previously and so the new aspect here is to analyse those data in a new way to look at different properties.

The paper is well structured, and the limited information in the data and methods section is mitigated by previous published work. Some reference to entrainment in stra-

tocumulus clouds specifically should be added.

The changes made from the original document have improved the manuscript, and it is much closer to publication. Where I still have comments or questions they are within the body of this report. The manuscript would still benefit from being more specific in places for clarity - some occasions identified in technical corrections.

**2 Specific Comments**

**2.1 Section 2**

I would like to see more information on interstitial aerosol observations. The size looks very large.

**2.2 Results Section 3.1**

It is interesting and somewhat unusual that the number concentrations increase with height above cloud base, rather than remaining relatively constant. I suggest noting this comparing to some of the VOCALS cloud observations perhaps.

**2.3 Results section 3.2**

Section 3.2, paragraph one. In the south eastern Pacific most of the aerosol optical depth will be within the marine boundary layer and so the assumption from the satellite studies is probably good here, as the aerosol and cloud layer are not well separated. Is there anything specific about the satellite studies that results in a large bias in this region? Otherwise it is not that relevant. Line 145 onwards: What altitude is the level

of decoupling in these clouds? Is it below the level where sub-CCN measurements are made? In the case of Nd and LWC, and cloud base even the "other" cases look well correlated apart from 2 - possibly the ones with precipitation? The decoupling will only have an impact if it is above the level where you make the sub-CCN measurements. Do you have measurements of the decoupling altitude?

2.4   Results Section 3.3

October 18th Case study: do all results here apply to this case? Is it possible to get aerosol particle size distribution for the sub-CCN layer, and the interstitial aerosol? It is a surprise that the unactivated aerosols are larger than 1 micron in size (for example in Figure 7. Is this because they are in a saturated environment? For example, during the VOCALS measurements (for example Twohy ACP2013, Impacts of aerosol particles on the microphysical and radiative properties of stratocumulus clouds over the southeast Pacific Ocean) observed much smaller interstitial aerosols of 150 nm, and below cloud 135 nm.

It looks as though the vertical velocity effect is limited for low total aerosol concentrations which seems interesting. Is this worth noting? Is the effect limited by low aerosol number?

Line 208. Is the average here for the whole flights worth of data for October 2018? Again - is it possible to show aerosol size distributions?

Why do some flights show a reduced effect, e.g. 22nd Oct, 29th, 30th, 4th Nov, 8th. Are the data able to explain?

**2.5   Results Section 3.4**

I still do not think there are strong difference in the vertical velocity PDFs between the well mixed and other cases. The grey shading does not help in figure 9, it might be easier to see if the shading is removed, and those cases are identified with a symbol above the axis. The standard deviations do not look different within the other category compared to well mixed, and if the skewness is not different, then what is? If anything I might expect the skewness to be the parameter that varied, when in a decoupled boundary layer, dominated by turbulence from cooling at cloud top, rather than the ocean surface thermals.

A see that the correlation reduces when the other cases are included, and so the dynamics are important (in Figure 9), but again - it looks like there are two strong outliers - which are these? Do they have to most skewed w PDFs or most different standard deviation of w? Or else precipitation, or wind shear?

**2.6   Results Section 3.5**

This section is interesting and appears to show some evidence for inhomogeneous mixing. It is difficult to isolate this, and I wonder if there is enough precision in the observations to look at 20 m deep layers. However the size distributions in Figure 11 show some reasonably convincing evidence. Does the degree of change in the size distribution correlate with the $AF_{ent}^{d}$ fraction in Table 2? For a quick look it appears to - is there a way to quantify this?

There are a number of references to entrainment in cumulus clouds, but these are not relevant here. The clouds are not still developing vertically at the inversion level, whereas in cumulus, at cloud top,the clouds are still growing. Lateral entrainment is important in cumulus, but not here.

Some reference include Malinkowski ACP2012 Physics of Stratocumulus Top (POST): turbulent mixing across capping inversion, Wood Monthly Weather Review 2012 Stratocumulus Clouds, and Stevens QJ2002, Entrainment in stratocumulus-topped mixed layers.

Line 285 - you suggest that entrainment of above cloud aerosol could be important, but elsewhere state it isn't, and showed this with the previous Figure 4. Line 287 - probability of what? Line 288 onwards - drier air would also case reduction in size. Line 312 - is this the increase in LWC from increased sub-CCN?

line 325 - do dynamical considerations mask the dispersion effect or is the effect lower once vertical velocity is considered? Line 334, 335 - the stratocumulus entrainment references may assist here. At cloud top vertical velocities will tend towards zero, and entrainment will dry the cloud and evaporate particles. There will not be much cloud nucleation here.

**3  Technical corrections**

There are numerous errors of tense and grammar that should be corrected.

Line 122, attributable Line 130, aerosols in, not on. Line 153, replace figure omitted with not shown line 163, As the certain... suggest re-writing for clarity line 164, replace contributed with controlled line 186, remove more, replace with spurious? As those extra aerosol area an artefact. line 196, Since part of.. suggest: Since part of the aerosol population has activated, or similar. line 200, and thus THEY activate line 209 Those aerosol, not that line 210 for INTO larger cloud droplets(Twohy

There are others to consider as well.

---

## Short Comment (SC1) · 30 Nov 2018

Comments on 2018 Atm. Chem. Phys. Discussions, paper #667:

We note that you define the entrainment and non-entrainment zone as the regions within 20-m above and below the height of maximum LWC in the VOCALS Sc, respectively; and you use these for describing the behavior of the droplet spectra and aerosol in the two zones. We also have been involved in an aircraft Sc field study (POST; 2008 off the CA coast) using the CIRPAS Twin Otter. The VOCALS and POST Sc should be quite similar given solid cloud cover and large temperature jumps above cloud top for both studies. Thus, it is of interest to do some initial comparisons here related to your comments on "entrainment in stratocumulus."

You suggest dry and warm air entrained in VOCALS Sc dilutes Nd and LWC but leaves droplet sizes relatively unaffected, thus resembling extreme inhomogenous entrainment/mixing. Also, you note that it is still unclear whether inhomogeneous or homogenous mixing dominates, and that previous studies favor inhomogenous mixing.

These suggestions are not far from what we found for the entrainment-mixing process in POST Sc; however, there are differences resulting from our look at the POST Sc using a different approach. We use a different vertical description in characterizing the region near Sc top (see Malinowski et al., 2013): "Cloud top" is defined as the maximum height of unbroken cloud. And above this cloud top the turbulent EIL (entrainment interface layer) extends up to the unaffected free atmosphere. Part of the EIL contains cloud filaments that are evaporating and where mixing can be termed extreme inhomogenous. Cloud and sensible heat detraining from cloud top are key in reducing buoyancy in the EIL. Radiative cooling is centered at cloud top causing the generation of negative buoyancy.

Then, what is entrained into the unbroken Sc below our cloud top, is it homogeneous or inhomogeneous? An answer is suggested by Fig. 1 which shows a 2-s (~100 m) data sample of LWC and Re (effective radius) collected on POST flight TO6.

[Figure]

Fig. 1 – LWC and Re (effective radius) 20-cm horizontal resolution

Sc data collected just below cloud top from POST flight TO6. Data

points indicate the location of a descending entrained cloud parcel.

The red data points indicate where both LWC and Re have decreased.

The LWC and Re data records in Fig. 1 both show relatively unchanging background values except for the location of the entrained parcel. Some Re values (red) show reduced values from the background indicating a change in the droplet size distribution and the probable presence of homogeneous mixing. However, most data points (black) only show a decrease in LWC. That sure looks again like extreme inhomogeneous mixing, but the black data more likely show a simple dilution of the cloud by air that has nearly the same buoyancy and moisture as cloud top, and that also preserves the value of Re. Gerber et al. (2016) give more details to support this conclusion including showing the independence of temperature reduction in entrained parcels from reduced LWC in the parcels (see Fig. 5 in that paper), and showing similar behavior of POST Sc with and without predicted buoyancy reversal using mixing fraction analysis.

Thus entrainment/mixing for the POST Sc, and likely also for the VOCALS Sc, is not simply extreme inhomogeneous mixing with warmer and dryer air causing the reduction of LWC and Nd in the Sc, but is a bit more complex with both extreme inhomogeneous mixing and homogenous mixing playing a role in the entrainment process. Yes, air entrained through Sc cloud top can change the droplet size spectrum as Fig. 1 suggests by the reduction of Re, but that appears secondary in most POST Sc in comparison to the dilution process that preserves the relative shape of the spectrum.

It would be interesting to know how your choice of entrainment and non-entrainment zones apply to the vertical-layer partitioning according to Malinowski et al. (2013), including your findings about droplet spectra and aerosol.

Malinowski, S.P., and Coauthors: Physics of Stratocumulus Top (POST): Turbulent mixing across capping inversion. Atmos. Chem. Phys., 13, 15233-15269, doi:10.5194/acpd-13-15223-2013, 2013.

Gerber, H., Malinowski, S.P., and Jonsson, H.: Evaporative and radiative cooling in POST stratocumulus. J. Atmos. Sci., 73, 3877-3884, doi:10.1175/JAS-D-16-0023.1, 2016.

---

## Author Comment (AC1) · 16 Feb 2019

Enclosed please find the point-by-point responses to the Referee #1's comments.

Please also note the supplement to this comment:
https://www.atmos-chem-phys-discuss.net/acp-2018-667/acp-2018-667-AC1-supplement.pdf

---

## Author Comment (AC2) · 16 Feb 2019

Enclosed please find the point-by-point responses to the short comments

Please also note the supplement to this comment:
https://www.atmos-chem-phys-discuss.net/acp-2018-667/acp-2018-667-AC2-supplement.pdf

---

## Referee Comment (RC2) · Anonymous Referee #3 · 19 Mar 2019

Review of "Exploring aerosol cloud interaction using VOCALS-REx aircraft measurements" by Jia et al.

The manuscript is a rexamination of the VOCALS aerosol-cloud dataset obtained from sixteen flights of the CIRPAS Twin Otter that each profiled the below-, in-, and above-cloud environment over the southeast Pacific Ocean. Relationships between the cloud droplet number and relative dispersion to sub-cloud CCN(s=0.2

1) The authors need to do a better job of explaining what this study contributes over and above the previous papers that have been published on VOCALS. Who is the audience for the paper (i.e., who would be interested in the findings)? How does the manuscript represent a substantial contribution to scientific progress (through substantially new concepts, ideas, methods, or data) as required by the ACPD publication criteria? Right

now, I would say that it does not represent a substantial contribution (which, in my opinion, makes this paper a borderline reject). Please discuss these details briefly in the abstract and more thoroughly in the last paragraph of the introduction; the current brief paper section layout discussion on Lines 62-64 is not particularly useful and could be replaced.

2) I have a couple of concerns about the treatment of the interstial aerosol. First, the PCASP-100 misses the large fraction of sub-0.1-um aerosols that are unlikely to act as CCN and would therefore remain as intersitial aerosols. Are there data from in-cabin particle counters sampling on an aerosol inlet that could fill in this major gap? Inlet shatter may be more of an issue here, but the sub-cloud measurements would be a good place to quantify the fraction of aerosol number that is sub- and super-0.1-um diameter. It is likely that many (or most) of the interstitial aerosol number is not being captured here.

3) The use of effective diameters to characterize the interstial aerosol (Di) and cloud droplet sizes (De) doesn't make sense to me as this paper is largely focused on number concentrations and number size distributions. While the effective diameter is relevant for remote sensing measurements, there are no remote sensing data presented in this paper. The authors should instead use geometric mean diameters to describe these aerosol populations and better convey the aerosol and cloud diameters relevant for the number distributions.

4) In a number of instances associations between sub-cloud and in-cloud variables are misinterpreted to suggest causal relationships that are inconsistent with our understanding of cloud physics. For exampele, on Lines 13-15, it is stated "Our analysis suggest (sic) that the increase in liquid water content (LWC) is mainly contributed by cloud droplet number concentration (Nd) instead of effective radius of cloud droplets in the polluted case, in which more droplets form with smaller size, while the opposite is true in the clean case." On Lines 142-144, it is stated: "This may imply that the increase of LWC induced by sub-CCN is mainly caused by increasing Nd instead of Re.

Fig. 3d indicates a positive correlation between cloud depth and sub-CCN..." These statements are either misleading or just not correct. LWC is known to be driven by changes in environmental conditions (i.e., profiles of temperature and total water content as well as entrainment mixing); microphysics are not a primary driver. Similarly, changes in these environmental conditions will also change the cloud base altitude (and hence cloud depth if the cloud top is driven by a constant inversion height). There is a causal link between sub-CCN and droplet number, while the in cloud supersaturation (again driven by environmental conditions) can also affect Nd. What the analysis shown in Figure 3 does suggest is that there is a correlation between higher sub-CCN loadings and wetter (or colder) environmental conditions, which should be discussed. The old axiom that correlation does not imply causation certainly holds here. These conclusions (on the lines cited above and elsewhere in the manuscript) need to be either revised or removed from the manuscript.

Minor Comments:

1) On Line 76, it is stated that the PCASP-100 measures the aerosol dry diameter. How was this accomplished? Was some sort of unique inlet heater or dryer used to dry the aerosol? While there will be some ram heating effects that will lower the relative humidity in the PCASP-100 optics region, I don't think that this would be enough to say that the aerosol size is dry.

2) Too many significant figures reported on Line 158. Reporting aerosol concentrations as integer values would be appropriate.

3) The manuscript would benefit from some additional proofreading to improve grammatical or typo errors.

---

## Author Comment (AC3) · 17 Apr 2019

**Response to Referee #3**

Thanks to the reviewer for the very helpful suggestions, which have allowed us to clarify and improve the manuscript. Below, we address the reviewer's comments, with the reviewer comments in black and our responses in blue. We have also revised the manuscript accordingly.

The manuscript is a rexamination of the VOCALS aerosol-cloud dataset obtained from sixteen flights of the CIRPAS Twin Otter that each profiled the below-, in-, and above- cloud environment over the southeast Pacific Ocean. Relationships between the cloud droplet number and relative dispersion to sub-cloud CCN(s=0.2

1) The authors need to do a better job of explaining what this study contributes over and above the previous papers that have been published on VOCALS. Who is the audience for the paper (i.e., who would be interested in the findings)? How does the manuscript represent a substantial contribution to scientific progress (through substantially new concepts, ideas, methods, or data) as required by the ACPD publication criteria? Right now, I would say that it does not represent a substantial contribution (which, in my opinion, makes this paper a borderline reject). Please discuss these details briefly in the abstract and more thoroughly in the last paragraph of the introduction; the current brief paper section layout discussion on Lines 62-64 is not particularly useful and could be replaced.

Thanks to the reviewer for the thoughtful comments and suggestions. The aerosol climatic effect is one of the greatest uncertainties in climate predictions, particularly the indirect effect, i.e., aerosol-cloud interaction. Data analyses based on ground, aircraft, and satellite measurements, as well as numerical simulations, are usually conducted to investigate aerosol-cloud interactions. Aircraft measurements from VOCALS provide detailed information on the microphysical properties of aerosols and clouds and their vertical profiles; thus, these measurements are very useful in studies on the topic discussed in this manuscript. We agree that there are quite a few previous studies that have been published on VOCALS measurements. Most of these studies either described the instruments or presented the properties of aerosols and clouds, which certainly improved our understanding of the properties of aerosols, clouds and BLs over the SEP. However, few of these previous studies explored the detailed processes of aerosol-cloud interactions, especially physical processes such as dispersion effects, entrainment mixing, and the impacts of these effects on the interaction. Using the microphysical properties of both aerosols and clouds, as well as the meteorological parameters from VOCALS, we explored the detailed processes on aerosol-cloud interaction over the SEP, including (a) the controlling factors of cloud droplet formation, (b) the dispersion effect after constraining the differences in cloud dynamics, and (c) the entrainment mixing process near the stratocumulus top and its impact on clouds. To our knowledge, such kind of analysis has not yet conducted by previous studies using VOCALS measurements.

The effect of aerosols on stratocumulus clouds is complicated by various dynamical conditions, e.g., strong wind shear within the BL, moist layers above clouds, and a strong decoupled BL. Thus,

we investigate the contribution of cloud dynamics and aerosols to cloud droplet formation, which to our knowledge has not received much attention in previous studies using VOCALS measurements.

In addition to modulating the cloud droplet number, aerosols can also change the shape of the cloud droplet size distribution (i.e., dispersion effect) and thereby cloud albedo (Liu and Daum, 2002). The dispersion effect could act to either offset or enhance the well-known Twomey effect, which mainly depends on the sensitivity of the relative dispersion ($\varepsilon$) to the aerosol number concentration ($Na$). However, as shown in Table 1, the relationship between $\varepsilon$ and $Na$ derived from previous studies remains largely uncertain, implying that the effect of aerosols on $\varepsilon$ is often intertwined with the effects of other factors, especially cloud dynamic conditions. Thus, it is necessary to isolate the $\varepsilon$ response to aerosol perturbations from meteorological effects, which to our knowledge, was not considered in many previous studies. A clear comparison between 'typical well-mixed' and 'other' cases in this manuscript can aid in understanding the influence of meteorological conditions on the dispersion effect estimation. Aerosols were found to broaden the droplet spectrum, and cloud dynamic perturbations may lead to an underestimation of the aerosol dispersion effect. This result is helpful for reducing the dispersion effect uncertainty and may benefit cloud parameterizations in global climate models to more accurately assess the indirect aerosol effect.

Additionally, entrainment plays a critical role in the formation and evolution of clouds and the change in droplet spectrum, as well as aerosol indirect effects (Chen et al., 2014, 2015; Andersen and Cermak, 2015). However, it remains unclear whether the entrainment-mixing mechanism is predominantly homogeneous, inhomogeneous, or in between (Andrejczuk et al., 2009; Lehmann et al., 2009). To our knowledge, little attention has been given to the entrainment-mixing mechanism obtained during VOCALS. Using cloud observations obtained from other aircraft (G-1) during VOCALS, Yum et al. (2015) showed that both homogeneous and inhomogeneous mixing were found in their analysis and attributed it to method uncertainties. We used a completely different method to re-examine the entrainment-mixing mechanism near the stratocumulus top. As stated by Gerber et al. (2005), in marine stratocumulus clouds, entrainment occurs when the LWC begins to decrease from the bottom of the cloud. In this manuscript, entrainment and non-entrainment zones are thus defined as the regions within 20 m above and below the maximal LWC height, respectively. A comparative analysis of the difference in cloud microphysics between the two zones suggests that the entrainment-mixing mechanism is predominantly extreme inhomogeneous in the stratocumulus clouds during VOCALS. The impacts of entrainment on cloud microphysics are also investigated. Previous studies have noted that applying different assumptions to the entrainment-mixing mechanism would have a significant impact on cloud albedo (Grabowski, 2006; Chosson et al., 2007; Slawinska et al., 2008). Therefore, our results provide insights to improve the understanding of entrainment mixing in stratocumulus clouds and the assessment of aerosol indirect effects and cloud radiative forcing.

According to the reviewer's suggestions, we have added the corresponding discussions to the abstract and introduction to ensure clarity of the novelty and contribution of this study, and we removed the paper section layout discussion from the text.

2) I have a couple of concerns about the treatment of the interstial aerosol. First, the PCASP-100 misses the large fraction of sub-0.1-um aerosols that are unlikely to act as CCN and would therefore remain as interstitial aerosols. Are there data from in-cabin particle counters sampling on an aerosol inlet that could fill in this major gap? Inlet shatter may be more of an issue here, but the sub-cloud measurements would be a good place to quantify the fraction of aerosol number that is sub- and super-0.1-um diameter. It is likely that many (or most) of the interstitial aerosol number is not being captured here.

We agree that the concentration of interstitial aerosols (> 0.1 μm) measured by PCASP-100 is less than the concentration of all un-activated aerosols, i.e., the interstitial aerosols with diameters smaller than 0.1 μm are not captured here. However, because in-cloud sampling of this part of the aerosols is problematic due to cloud droplet shatter (Hudson and Frisbie, 1991; Clarke et al., 1997; Weber et al., 1998; Kleinman et al., 2012), we do not include it into interstitial aerosols in this manuscript. As suggested by Kleinman et al. (2012), cloud droplet shatter can create a large number of spurious small particles (~ 50 nm), leading to a serious overestimation of interstitial aerosol concentration. Additionally, other previous studies also observed extremely high concentrations of these small particles in-cloud (e.g., $10^3$ to more than $10^4$ particles $cm^{-3}$ smaller than 50 nm diameter) and attributed these concentrations to droplet shatter (Hudson and Frisbie, 1991; Clarke et al., 1997; Weber et al., 1998). However, the shatter contribution to the total in-cloud aerosols is minor when the diameter is greater than 0.1-0.15 μm (Kleinman et al., 2012). Therefore, treating aerosols larger than 0.1 μm as interstitial aerosols can avoid the interference of cloud droplet shatter to a large extent, which is precisely what we did in this manuscript.

According to the reviewer's suggestions, we calculated the ratio of the sub-0.1-μm aerosol concentration to the super-0.1-μm aerosol concentration during flights on Oct. 18, where sub-0.1-μm aerosol concentration is derived from the concentration measured by CPC (size range: > 15 nm) minus that measured by PCASP-100 (size range: 0.1-2.0 μm), and the super-0.1-μm aerosol concentration is obtained from the concentration measured by PCASP-100. It is found that the average ratio for in-cloud (2.99) is significantly higher than that for sub-cloud (0.31), which further confirms the contribution of droplet shatter to the spurious increase in small aerosols in clouds. Notably, the average ratio for sub-cloud is only 0.31, indicating that some aerosols could be missed by PCASP-100, but these aerosols are not the majority. Furthermore, as the reviewer stated, most aerosols smaller than 0.1 μm are unlikely to act as CCN and thus affect cloud properties. Therefore, due to the weak connection with CCN, these aerosols that are missed by PCASP-100 may not be important when exploring the impact of aerosols on clouds. For example, the same treatment of interstitial aerosols as in this manuscript has also been employed by other studies (Kleinman et al., 2012) to examine the aerosol effect on clouds.

3) The use of effective diameters to characterize the interstial aerosol (Di) and cloud droplet sizes (De) doesn't make sense to me as this paper is largely focused on number concentrations and number size distributions. While the effective diameter is relevant for remote sensing measurements, there are no remote sensing data presented in this paper. The authors should instead use geometric mean diameters to describe these aerosol populations and better convey the aerosol and cloud diameters relevant for the number distributions.

Thanks for the comments. Our focus in this manuscript is to investigate the effect of aerosols on clouds and hence provide a reference for the assessment of aerosol indirect radiative forcing. The climatic effects of aerosols and clouds are usually estimated based on the radiatively important effective diameters rather than the geometric mean diameters. For this reason, many aircraft-based studies on aerosol-cloud interactions also used effective diameters to represent the sizes of aerosols and cloud droplets (Peng et al., 2002; Zhang et al., 2011; Vogelmann et al., 2012; Yang et al., 2019; Zhao et al., 2018, 2019), although there was no comparison with the remote sensing measurements. Additionally, effective diameters are often used to quantify the extent of dilution of the cloud caused by the entrainment-mixing process (Pontikis et al., 1993; Gerber et al., 2008, 2013; 2016). For the above reasons, we used effective diameters instead of geometric mean diameters in this manuscript.

We agree with the reviewer that the geometric mean diameter is more relevant for the number size distributions. Thus, we compared the effective diameters with the geometric mean diameters of aerosols and cloud droplets, respectively (Figure R1), and found that there is a good correlation between them. This correlation implies that using the two different diameters would not influence our conclusions.

[Figure]

Figure R1. Correlation between geometric mean diameters and effective diameters of (a) aerosols and (b) cloud droplets, respectively. Colors represent the frequency (units: %).

4) In a number of instances associations between sub-cloud and in-cloud variables are misinterpreted to suggest causal relationships that are inconsistent with our understanding of cloud physics. For exampele, on Lines 13-15, it is stated "Our analysis suggest (sic) that the increase in liquid water content (LWC) is mainly contributed by cloud droplet number concentration (Nd) instead of effective radius of cloud droplets in the polluted case, in which more droplets form with smaller size, while the opposite is true in the clean case." On Lines 142-144, it is stated: "This may imply that the increase of LWC induced by sub-CCN is mainly caused by increasing Nd instead of Re. Fig. 3d indicates a positive correlation between cloud depth and sub-CCN..." These statements are either misleading or just not correct. LWC is known to be driven by changes in environmental conditions (i.e., profiles of temperature and total water content as well as entrainment mixing); microphysics are not a primary driver. Similarly, changes in these environmental conditions will also change the cloud base altitude (and hence cloud depth if the cloud top is driven by a constant

inversion height). There is a causal link between sub-CCN and droplet number, while the in cloud supersaturation (again driven by environmental conditions) can also affect Nd. What the analysis shown in Figure 3 does suggest is that there is a correlation between higher sub-CCN loadings and wetter (or colder) environmental conditions, which should be discussed. The old axiom that correlation does not imply causation certainly holds here. These conclusions (on the lines cited above and elsewhere in the manuscript) need to be either revised or removed from the manuscript.

Thanks for the comments. We agree with the reviewer that both environmental conditions and microphysics (i.e., aerosol effects) can affect cloud properties, and usually, the former is the primary driver. On this point, we distinguished between the flights of the typical mixed BL and others to ensure relatively similar meteorological conditions (similar inversion heights, and the jump of potential temperature and total water mixing ratio across the inversion). In addition, the in-cloud dynamics (i.e., vertical velocity) for the 16 non-drizzling flights were also compared (Figure 9 and Table 2). The result indicated that the in-cloud dynamic differences between the typical well-mixed boundary flights is very small, which confirms the assumption of similar meteorological conditions. Therefore, the interference of environmental conditions on the relationships between the cloud and sub-CCN shown in Figure 3 would be minimal.

LWC is a function of both the number (Nd) and size (Re) of cloud droplets, i.e., both Nd and Re contribute to the change in LWC. The objective of the analysis on the relationships between LWC and Nd/Re (Lines 13-15; Figure 4) is to understand which contributed more to the growth of LWC under polluted and clean conditions, i.e., cloud formation under different aerosol loadings. Our analysis shows that the low aerosol concentrations in the clean case inhibit the increase in Nd with LWC, which promotes the rapid increase in Re with LWC. In contrast, there are enough particles that may potentially be activated into cloud droplets under polluted conditions; thus, Nd increases rapidly with LWC, while the increase in Re is suppressed. We agree with the reviewer that the sentences on lines 13-15 and lines 142-144 might be misunderstood, and thus, these sentences have been removed from revised manuscript as suggested.

Minor Comments:

1) On Line 76, it is stated that the PCASP-100 measures the aerosol dry diameter. How was this accomplished? Was some sort of unique inlet heater or dryer used to dry the aerosol? While there will be some ram heating effects that will lower the relative humidity in the PCASP-100 optics region, I don't think that this would be enough to say that the aerosol size is dry.

Thanks for this comment. The relative humidity of the air sampled by PCASP can be reduced to 40 % by using deicing heaters, which increases the temperature by approximately 10 to 20 °C (Strapp et al., 1992; Hallar et al., 2006; Snider and Petters, 2008). Cabin heat also contributed to the drying of aerosols measured inside the aircraft. We agree with the reviewer that this may be not enough to say the diameter is dry, although the relative humidity is already very low. Thus, we have revised "dry diameter" to "diameter" on line 104.

2) Too many significant figures reported on Line 158. Reporting aerosol concentration as integer

values would be appropriate.

Thanks for the suggestion. In the revised manuscript, the number concentrations of aerosols and cloud droplets were reported as integer values throughout the paper.

3) The manuscript would benefit from some additional proofreading to improve grammatical or typo errors.

We have carefully proofread the manuscript and asked a native English speaker to read and edit the language of the manuscript.

[revised manuscript text omitted]

Fig. 2 panels showing vertical profiles. Top row axes: (a) T (°C), (b) RH (%), (c) $N_d$ (#/cc), (d) LWC (g m$^{-3}$), (e) $R_e$ (µm), (f) $D_a$ (µm), (g) CCN/CN (SS=0.2%). Bottom row axes: (a) T (°C), (b) RH (%), (c) LWC (g m$^{-3}$), (d) $R_e$ (µm), (e) $N_d$ (#/cc), (f) $R_a$ (µm), (g) CCN/CN (SS=0.2%). Vertical axis: $z/z_i$.

**Fig. 2. Vertical profiles scaled by the inversion height. (a) temperature (K); (b) relative humidity (%); (c)** liquid water content (g m

$^{-3}$)**; (d)** cloud droplet effective radius (µm) **; (e)** cloud droplet

**number concentration (cm $^{-3}$); (f)  aerosols effective radius (μm), and (g) the number concentration ratio of CCN to aerosols for all 16 non-drizzling flights. The gray lines show all individual flights, and the orange lines indicate the average profiles. The red and green lines represent the polluted (Oct. 18) and clean (Nov. 9) cases, respectively.**

[Figure]

**Fig. 3. (a) *LWC* (g cm $^{-3}$); (b) $N_d$ (cm $^{-3}$); (c) $R_e$ (μm) as a function of sub-cloud CCN concentrations (SS=0.2%) for all 16 non-drizzling flights. The error bars through these symbols indicate the standard deviation. Red symbols are the cases with typical well-mixed BL  discussed in Zheng et al. (2011), and blue symbols for others. Red (black) texts are the correlation coefficient for typical well-mixed cases (all cases).**

[Figure]

740 Fig. 4. Correlations between (a) $N_d$ (cm$^{-3}$), (b) $R_e$ (μm) and $LWC$ (g m$^{-3}$) for clean (green) and polluted (red) cases, respectively.

[Figure]

745 Fig. 5. Vertical profiles of number concentrations of aerosols ($N_a$), cloud droplets ($N_d$) and total in-cloud particles ($N_d + N_i$) during

the flight on Oct. 18.

[Figure]

**Fig. 6. Relationships between** $N_d$ **and** $N_i + N_d$ **during all 16 non-drizzling flights. The colors represents in-cloud vertical velocities (m**

s$^{-1}$**), and gray line is 1:1 line. The mean and standard deviation of** $N_d/(N_d+N_i)$ **for vertical velocity greater than 1 m s$^{-1}$ (red) and less**

**than -1 m s$^{-1}$ (blue) are shown.**

[Figure]

755 **Fig. 7. Same as Fig. 6, but the colors represents the effective  radius of interstitial aerosol ($\cancel{D_i}R_i$) (µm). The mean and standard deviation of $N_d/(N_d+N_i)$ for $R_i$ greater than 0.5 µm (red) and less than 0.25 µm (blue) are shown.**

[Figure]

**Fig. 8.** Relationships between relative dispersion (ε) and $N_d$ during all 16 non-drizzling flights, in which the colors  **represents** in-cloud vertical velocities (m s$^{-1}$).

760

[Figure]

[Figure]

**Fig. 9.** Probability distribution function (units: %) of vertical velocity (*w*) for 16 non-drizzling flights. Black symbols are mean values of *w*, and error bars through these symbols indicate the standard deviation. Circles are the cases with typical well-mixed BL, and crosses represents the other cases.

[Figure]

**Fig. 10.** Relative dispersion (*ε*) as a function of (a) $N_d$ and (b) sub-cloud CCN concentrations (*SS*=0.2%) for all flights. The error bars through these symbols indicate the standard deviation. Red symbols are the cases with typical  well-mixed BL , and blue symbols for others. Red (black) texts are the correlation coefficient and slope for typical well-mixed cases (all cases).

[Figure]

**Fig. 11.** Number size distributions of cloud droplets in the entrainment (yellow) and non-entrainment zones (blue) during all 16 non-drizzling flights.

[Figure]

[Figure]

**Fig. 12.** Probability density functions of (a)  $R_i$ (µm), (b) $RH$ (%), (c) $N_d/(N_d + N_i)$, and (d) $\varepsilon$ in the entrainment (yellow) and non-entrainment zones (blue) during the flight on Oct. 18.

*Supplement of*

**Exploring aerosol cloud interaction using VOCALS-REx aircraft measurements**

Hailing Jia, Xiaoyan Ma and Yangang Liu

*Correspondence to*: Xiaoyan Ma (xma@nuist.edu.cn)

**Figure List**

Figure S1. Normalized profiles of $N_d$. Values of $Z_N=0$ indicates the cloud base whereas $Z_N=1$ the cloud top. Orange line indicates the average profiles.

Figure S2. (a) $P_{LWC}$ and (b) $P_{Nd}$ as a function of $AF_{ent}/AF_{non-ent}$ for all 16 non-drizzling flights.

**Figure S1**

[Figure]

**Figure S2**

[Figure]

---

## Author Response (AR2)

Dear Co-Editor,

We thank the reviewers for their thoughtful comments, which have allowed us to clarify and improve the manuscript. We have carefully revised the manuscript according to the reviewer's suggestions and comments.

In particular, we have made substantial changes to the abstract and introduction to clarify the novelty and contribution of this study. Although previous studies published on VOCALS measurements have certainly improved our understanding of some aspects related to aerosols, clouds and boundary layer (BL) properties over the southeast Pacific Ocean (SEP), several important factors remain understudied or unexplored. First, the effect of aerosols on clouds is often intertwined with the effects of other factors, especially meteorological conditions. Currently, the impact of aerosols on the shape of the cloud droplet size spectrum (i.e., dispersion effect) is reported to remain largely uncertain, which may be mostly attributable to the coincidentally changing cloud dynamics. Thus, it is necessary to isolate the response of relative dispersion to aerosol perturbations from meteorological effects, which to our knowledge, has not been the focus of attention in many previous studies. Second, applying different assumptions to the entrainment-mixing mechanism has a significant impact on the cloud albedo. However, it remains unclear whether the entrainment-mixing mechanism is predominantly homogeneous, inhomogeneous, or in between, which could lead to inaccurate assessments of aerosol indirect effects. Thus, more attention should be given to this topic.

In this manuscript, we examine the detailed processes of aerosol-cloud interactions over the SEP, with a focus on the understudied topics (separation of aerosol effects from dynamic effects, dispersion effects, and turbulent entrainment-mixing processes). This study is helpful for reducing the uncertainty in dispersion effects and entrainment mixing in stratocumulus clouds, and the result of this study may benefit cloud parameterizations in global climate models to more accurately assess the aerosol indirect effects.

Listed below are our responses to the comments from the reviewers. For clarity and visual distinction, the reviewer's comments are listed in black and the authors' responses are in blue.

Sincerely,

Xiaoyan Ma, Hailing Jia, and Yangang Liu

**Response to Referee #1**

**1 Overview Comments**

This paper uses data from a field campaign in the south eastern Pacific to investigate the aerosol dispersion effect and entrainment in stratocumulus clouds. The cases have been described in other work previously and so the new aspect here is to analyse those data in a new way to look at different properties.

The paper is well structured, and the limited information in the data and methods section is mitigated by previous published work. Some reference to entrainment in stratocumulus clouds specifically should be added.

The changes made from the original document have improved the manuscript, and it is much closer to publication. Where I still have comments or questions they are within the body of this report. The manuscript would still benefit from being more specific in places for clarity - some occasions identified in technical corrections.

We thank the reviewer for taking the time to assess the manuscript and for providing helpful comments and suggestions to improve the manuscript. We have revised the manuscript carefully according to the reviewer's comments. At the same time, we are grateful for the important references provided by the reviewer. These and other references related to entrainment in stratocumulus have been cited in the revised manuscript. Please see the following detailed point-by-point responses.

**2 Specific Comments**

2.1 Section 2

I would like to see more information on interstitial aerosol observations. The size looks very large.

In this study, the size distribution of interstitial aerosol is obtained directly from the observation of in-cloud aerosols by Passive Cavity Aerosol Spectrometer Probe (PCASP-100), which counted and sized particles from 0.1–2.0 μm dry diameter with 20 bins. The description has been added in section 2.1 accordingly (line 103 in the revised manuscript). For an explanation of the large size, please see the detailed responses to section 2.4 (Results Section 3.3).

2.2 Results Section 3.1

It is interesting and somewhat unusual that the number concentrations increase with height above cloud base, rather than remaining relatively constant. I suggest noting this comparing to some of the VOCALS cloud observations perhaps.

Thanks for reminder. We agree that, in most cases, $N_d$ profiles should be close to relatively constant,

but this is not always the case (Keil et al., 2003). We realize that the normalization by cloud-top height only may be insufficient to indicate the vertical variation of clouds when all profiles are averaged, because each profile has a different cloud base. Thus, the average profiles are removed from the Fig. 2c, 2d, 2e, and the vertical variation of cloud properties can be seen easily from the single profile. As depicted in Fig. 2c, the green $N_d$ profile remains relatively constant, and the red one shows a slight increase with height. Furthermore, to get the average profile of all flights reasonably, we normalize the height $Z_N=(Z-Z_{base})/\Delta Z$, where $Z_{base}$ and $\Delta Z$ are the cloud base height and the geometrical cloud depth, respectively (Fig. R1). This transformation implies that $Z_N=1$ at the cloud top, and $Z_N=0$ at the cloud base. As shown in Fig. R1, the average profile of Nd remains relatively constant with a slight increase and decrease near base and top respectively, which is consistent with results in other VOCALS-REx observations (Painemal and Zuidema, 2011). We have modified the main text related to $N_d$ profile accordingly (lines 154-160 in the revised manuscript).

[Figure]

Figure R1. Normalized profiles of $N_d$. Values of $Z_N=0$ indicates the cloud base whereas $Z_N=1$ the cloud top. Orange line indicates the average profiles.

2.3 Results section 3.2

Section 3.2, paragraph one. In the south eastern Pacific most of the aerosol optical depth will be within the marine boundary layer and so the assumption from the satellite studies is probably good here, as the aerosol and cloud layer are not well separated. Is there anything specific about the satellite studies that results in a large bias in this region? Otherwise it is not that relevant.

As reported in previous studies (Allen et al., 2011; Shank et al., 2012), biomass burning serves as a potential source of aerosol to the free troposphere above cloud over the South East Pacific (SEP) region. Under the influence of biomass burning plume which carry elevated organic combustion aerosol, the aerosol concentration above cloud becomes comparable to that below cloud (Allen et al., 2011). By using satellite data, Costantino et al.(2010) pointed out that aerosols from biomass burning are often separated from the underlying stratocumulus cloud layers, and thus have little effect on cloud properties. Therefore, in this case, AOD as a proxy of CCN number concentration to investigate the aerosol-cloud interactions could induce biases. It is necessary to investigate the impact of CCN number

concentration near cloud layer on cloud properties.

Line 145 onwards: What altitude is the level of decoupling in these clouds? Is it below the level where sub-CCN measurements are made? In the case of Nd and LWC, and cloud base even the "other" cases look well correlated apart from 2 - possibly the ones with precipitation? The decoupling will only have an impact if it is above the level where you make the sub-CCN measurements. Do you have measurements of the decoupling altitude?

The decoupling is characterized by a vertically non-uniform distribution of total water mixing ratio from the surface to the capping inversion, or a cumulus cloud underlying stratocumulus (Zheng et al., 2018). Based on this, we derived the decoupling altitude (Table R1). The two outliers are 1[st] Nov (drizzling case) and 29[th] Oct (decoupling case). As shown in Table R1, the decoupling height for 29[th] Oct is indeed above the level where the sub-CCN measurements were made.

Furthermore, we have removed the drizzling cases from Fig. 3 in revised manuscript, and reanalyzed the relationships between sub-CCN and cloud properties for all flight and well mixing flights, respectively. It is found that the correlation coefficients between sub-CCN and LWC (Fig. 3a) and cloud base height (Fig. 3f) for all non-drizzling flights are 0.38 and -0.52, respectively, which are significantly lower than those for well mixing cases (0.60 and -0.69), confirming that the aerosol effect could be confounded by various dynamics. However, the change in correlation coefficient between sub-CCN and Nd is very small (0.83 vs. 0.79). One possible explanation is that, the impact of aerosol on Nd is relatively linear and direct, while LWP is a function of both Re and Nd, which depends not only on the number of condensation nuclei, but also on the subsequent growth process of cloud droplets, and thus is more sensitive to dynamics. Similarly, the relative dispersion is also strongly dependent on dynamics (Fig. 8). Therefore, even if Nd does not show a clear difference between the well mixing and other cases, it is still necessary to distinguish meteorological categories. In this study, all 'other' cases that could confuse the aerosol effect are eliminated, such as decoupling and wind shear, which affect the feeding of water vapor and energy from the surface.

Table R1. The heights of decoupling and cloud base for three decoupling cases.

| Date | 10.29 | 11.04 | 11.08 |
|---|---|---|---|
| Decoupling Height (m) | 810.3 | 631.7 | 844.2 |
| Cloud Base Height(m) | 850.4 | 920.5 | 1238.3 |

2.4 Results Section 3.3

October 18th Case study: do all results here apply to this case? Is it possible to get aerosol particle size

distribution for the sub-CCN layer, and the interstitial aerosol? It is a surprise that the unactivated aerosols are larger than 1 micron in size (for example in Figure 7. Is this because they are in a saturated environment? For example, during the VOCALS measurements (for example Twohy ACP2013, Impacts of aerosol particles on the microphysical and radiative properties of stratocumulus clouds over the southeast Pacific Ocean) observed much smaller interstitial aerosols of 150 nm, and below cloud 135 nm.

In section 3.3, only Fig. 5 applies to 18th Oct case, and the rest of the results are for the average of all cases. This has been specified in the revised manuscript. For better understanding, the average Nd/(Nd+Ni) applied to different conditions for each individual flight have been also added to Fig. 6 and Fig. 7 in the revised manuscript.

The size distributions for the sub-cloud aerosol and the in-cloud (interstitial) aerosol are shown in Fig. R2. By directly comparing, it seems that the size of in-cloud aerosol in this study is larger than that in Twohy et al.(2013). However, it should be noted that the size shown in Fig. 7 is the individual sampling at specific locations in the cloud (instantaneous sampling), while the size of 150 nm in Twohy et al.(2013) is an average of a flight, where some large values might be smoothed. Another possible explanation is that, we use the effective diameter (Zhang et. al, 2011) to represent the aerosol size distribution rather than geometric mean diameter utilized in Twohy et al.(2013). For comparison purposes, the averaged geometric mean diameters of sub-cloud (blue, 184 nm) and in-cloud (red, 181 nm) aerosols during 18th Oct is also calculated (Fig. R2), which is much closer to the size in Twohy et al.(2013), but with a slight overestimation (~ 40 nm). This might be attributed to the difference in the measurement range of the instruments, i.e., 0.055–1.0 μm for Ultra High Sensitivity Aerosol Spectrometer (UHSAS) in Twohy et al.(2013), but 0.1–2.0 μm for Passive Cavity Aerosol Spectrometer Probe (PCASP-100) in our study. The latter is unable to observe the Aitken mode that is less than 0.1 μm, thus its geometric mean diameters is larger. In summary, comparing the aerosol in this study with that in Twohy et al.(2013) under the same conditions, the two are very close.

We agree that the aerosol size might be overestimated in a saturated environment (Fig. R2). Thus, in order to eliminate the influence of strong supersaturation on aerosol size, we exclude the samples with RH larger than 97%, and reanalyze the dependence of Nd/(Nd+Ni) on Di (Fig. R3). It is found that, without strong supersaturation, Nd/(Nd+Ni) still tend to increases with Di, so it seems saturated environment might not influence our conclusion significantly.

[Figure]

Figure R2. The size distributions for the sub-cloud aerosol (blue) and the in-cloud aerosol (red) during the flight on 18th Oct.

[Figure]

Figure R3. Relationships between Nd and Ni + Nd during all 16 non-drizzling flights when RH is larger than 97%. The colors represent the effective diameter of interstitial aerosol ($D_i$) (μm), and gray line is 1:1 line.

It looks as though the vertical velocity effect is limited for low total aerosol concentrations which seems interesting. Is this worth noting? Is the effect limited by low aerosol number?

To check if this limitation exists, we compared the difference of Nd/(Nd+Ni) between large and small vertical velocity for each flight (Fig. 6). It is found that there is no significant difference between low and high aerosol concentrations cases. Thus, this might be caused by visual effects, because in the case of low aerosol concentrations, most of the data concentrate and hence overlap each other in Figures.

Line 208. Is the average here for the whole flights worth of data for October 2018? Again - is it possible to show aerosol size distributions?

The Nd/(Nd+Ni) here is the average of all flights. The aerosol size distributions is shown in Fig. R2. For a more detailed discussion of aerosol size, please see the response to the previous section.

Why do some flights show a reduced effect, e.g. 22nd Oct, 29th, 30th, 4th Nov, 8th.   Are the data able to explain?

The Nd/(Nd+Ni) for each individual flight are calculated and shown in Fig. 7. It is demonstrated that these flights do not show a reduced effect.

2.5 Results Section 3.4

I still do not think there are strong difference in the vertical velocity PDFs between the well mixed and other cases. The grey shading does not help in figure 9, it might be easier to see if the shading is removed, and those cases are identified with a symbol above the axis. The standard deviations do not look different within the other category compared to well mixed, and if the skewness is not different, then what is? If anything I might expect the skewness to be the parameter that varied, when in a decoupled boundary layer, dominated by turbulence from cooling at cloud top, rather than the ocean surface thermals.

Thanks for suggestions. We have removed the shading from Fig. 9. The well-mixing and other cases are marked as circles and crosses, respectively. The means, standard deviations, and skewnesses of vertical velocities for all flights have been added in Table 2. Indeed, there is no significant difference in standard deviations between well mixing and other cases, but the means of other cases are overall smaller than that of well mixing cases. However, 4th Nov is an exception with a mean value close to well mixing cases, but its skewness is relatively large. That is, there are some differences in the vertical velocity between the well mixed and other cases (Table 2), implying the importance of distinguishing the well mixing cases from other cases.

A see that the correlation reduces when the other cases are included, and so the dynamics are important (in Figure 9), but again - it looks like there are two strong outliers - which are these? Do they have to most skewed w PDFs or most different standard deviation of w? Or else precipitation, or wind shear?

The two strong outliers in Fig. 10 are 24$^{th}$ Oct and 13$^{th}$ Nov, which are characterized by a strong wind shear (Fig. R4). For these two cases, the average in-cloud $w$ are smaller (-0.06 and -0.02) and the relative dispersions are larger (0.46 and 0.41), showing the dependence of relative dispersion on $w$ (as indicated in Fig. 8), which further highlights the importance of minimizing the influences of

meteorological conditions by excluding the other cases.

[Figure]

Figure R4. Vertical profiles of (a) horizontal wind speed and (b) wind direction during flights on 24[th] Oct (red) and 13[th] Nov. Dashed lines indicate the height of the cloud base.

2.6 Results Section 3.5

This section is interesting and appears to show some evidence for inhomogeneous mixing. It is difficult to isolate this, and I wonder if there is enough precision in the observations to look at 20 m deep layers. However the size distributions in Figure 11 show some reasonably convincing evidence. Does the degree of change in the size distribution correlate with the AFdent fraction in Table 2? For a quick look it appears to - is there a way to quantify this?

As shown in Fig. R5, the vertical speed of the CIRPAS Twin Otter aircraft ranges from -5 to 5 m s[-1], most of which are concentrated between -1 and 1 m s[-1]. Therefore, it is sufficient to observe the 20 m deep layers, especially during the horizontal legs near the cloud top where we distinguish the entrainment and the non-entrainment zone.

Thanks for suggestions. In order to check the relationship between the degree of change in the size distribution and adiabatic fraction, we correlated $AF_{ent}/AF_{non-ent}$ with $P_{LWC}$ and $P_{Nd}$, respectively, where $AF_{ent}/AF_{non-ent}$ indicates the change of AF in the entrainment zone relative to that in the non-entrainment zone (Fig. R6). It is shown that both $P_{LWC}$ and $P_{Nd}$ are negatively correlated with $AF_{ent}/AF_{non-ent}$, with correlation coefficients of -0.60 and -0.47, respectively, implying the dependence of the changes in the size distribution on the changes in adiabatic fraction. The result has been added in section 3.5 accordingly (line 346-349 in the revised manuscript).

[Figure]

Figure R5. Probability density functions of the vertical speed of the CIRPAS Twin Otter aircraft during the flight on 18[th] Oct.

[Figure]

Figure R6. (a) $P_{LWC}$ and (b) $P_{Nd}$ as a function of $AF_{ent}/AF_{non-ent}$ for all 16 non-drizzling flights.

There are a number of references to entrainment in cumulus clouds, but these are not relevant here. The clouds are not still developing vertically at the inversion level, whereas in cumulus, at cloud top, the clouds are still growing. Lateral entrainment is important in cumulus, but not here.

Some reference include Malinkowski ACP2012 Physics of Stratocumulus Top (POST): turbulent mixing across capping inversion, Wood Monthly Weather Review 2012 Stratocumulus Clouds, and Stevens QJ2002, Entrainment in stratocumulus-topped mixed layers.

Thanks very much for valuable suggestions. We agree that vertical velocities at the top of stratocumulus are much weaker than that of cumulus, and hence there might not be much cloud nucleation here. We have modified the text in section 3.5 accordingly, and those references have been also included to support the conclusion.

Line 285 - you suggest that entrainment of above cloud aerosol could be important, but elsewhere state it isn't, and showed this with the previous Figure 4.

We agree that entrainment of above cloud aerosol might be not important due to the negligible cloud nucleation here. Also, we have modified the text in section 3.5 accordingly.

Line 287 - probability of what?

It is the probability of Di.

Line 288 onwards - drier air would also case reduction in size.

In Figure. 2f, it is clearly shown that the effective diameter of aerosol particles above cloud is smaller than that below cloud. To minimize the effect of saturated environment on aerosol size, we excluded the data with relative humidity greater than 97%, and found that the aerosol size in the entrainment zone is still smaller than that in the non-entrainment zone. This result implies that small particles are indeed entrained into cloud from the top.

Line 312 - is this the increase in LWC from increased sub-CCN?

This part of the analysis is intended to illustrate the cloud formation in different aerosol loadings, i.e., for the polluted condition, the increase of LWC is mainly contributed by Nd instead of Re, in which large number of cloud droplets are formed with smaller size, and the reverse is true for clean the condition. Of course, this can also be used to support the conclusion that LWC increases with sub-CCN due to more cloud droplet.

line 325 - do dynamical considerations mask the dispersion effect or is the effect lower once vertical velocity is considered?

In general, the different dynamics mask the aerosol effect on relative dispersion. As indicated in Fig 10, if do not constrain the differences of cloud dynamics, the positive slope of aerosol concentration versus relative dispersion tends to be weaker, i.e., an underestimation of dispersion effect.

Line 334, 335 - the stratocumulus entrainment references may assist here. At cloud top vertical velocities will tend towards zero, and entrainment will dry the cloud and evaporate particles. There will not be much cloud nucleation here.

Thanks for suggestions. As reviewer stated, inversion capping a typical stratocumulus is usually too strong to excite significant updrafts near cloud top (Stevens, 2002; Wood, 2012; Malinowski et al., 2013). Ghate et al. (2010) found that vertical velocities near the top of stratocumulus overall tend towards zero with only about 4% of updrafts stronger than 0.5 m s$^{-1}$. Therefore, although smaller aerosols are entrained into the entrainment zone, these aerosols seem unlikely to influence droplet formation by inhibiting activation due to the negligible cloud nucleation here. The effect of entrainment mixing on stratocumulus is mainly governed by the entrained dry air rather than small aerosols. These

discussions have been included in section 3.5. The text has been revised accordingly.

**3 Technical corrections**

There are numerous errors of tense and grammar that should be corrected. Line 122, attributable Line 130, aerosols in, not on. Line 153, replace figure omitted with not shown line 163, As the certain... suggest re-writing for clarity line 164, replace contributed with controlled line 186, remove more, replace with spurious? As those extra aerosol area an artefact. line 196, Since part of.. suggest: Since part of the aerosol population has activated, or similar. line 200, and thus THEY activate line 209 Those aerosol, not that line 210 for INTO larger cloud droplets(Twohy

There are others to consider as well.

All revised. Thanks.

Thanks to the reviewer for the thoughtful comments and suggestions. The aerosol climatic effect is one of the greatest uncertainties in climate predictions, particularly the indirect effect, i.e., aerosol-cloud interaction. Data analyses based on ground, aircraft, and satellite measurements, as well as numerical simulations, are usually conducted to investigate aerosol-cloud interactions. Aircraft measurements from VOCALS provide detailed information on the microphysical properties of aerosols and clouds and their vertical profiles; thus, these measurements are very useful in studies on the topic discussed in this manuscript. We agree that there are quite a few previous studies that have been published on VOCALS measurements. Most of these studies either described the instruments or presented the properties of aerosols and clouds, which certainly improved our understanding of the properties of aerosols, clouds and BLs over the SEP. However, few of these previous studies explored the detailed processes of aerosol-cloud interactions, especially physical processes such as dispersion effects, entrainment mixing, and the impacts of these effects on the interaction. Using the microphysical properties of both aerosols and clouds, as well as the meteorological parameters from VOCALS, we explored the detailed processes on aerosol-cloud interaction over the SEP, including (a) the controlling factors of cloud droplet formation, (b) the dispersion effect after constraining the differences in cloud dynamics, and (c) the entrainment mixing process near the stratocumulus top and its impact on clouds. To our knowledge, such kind of analysis has not yet conducted by previous studies using VOCALS measurements.

The effect of aerosols on stratocumulus clouds is complicated by various dynamical conditions, e.g., strong wind shear within the BL, moist layers above clouds, and a strong decoupled BL. Thus,

we investigate the contribution of cloud dynamics and aerosols to cloud droplet formation, which to our knowledge has not received much attention in previous studies using VOCALS measurements.

In addition to modulating the cloud droplet number, aerosols can also change the shape of the cloud droplet size distribution (i.e., dispersion effect) and thereby cloud albedo (Liu and Daum, 2002). The dispersion effect could act to either offset or enhance the well-known Twomey effect, which mainly depends on the sensitivity of the relative dispersion ($\varepsilon$) to the aerosol number concentration ($N_a$). However, as shown in Table 1, the relationship between $\varepsilon$ and $N_a$ derived from previous studies remains largely uncertain, implying that the effect of aerosols on $\varepsilon$ is often intertwined with the effects of other factors, especially cloud dynamic conditions. Thus, it is necessary to isolate the $\varepsilon$ response to aerosol perturbations from meteorological effects, which to our knowledge, was not considered in many previous studies. A clear comparison between 'typical well-mixed' and 'other' cases in this manuscript can aid in understanding the influence of meteorological conditions on the dispersion effect estimation. Aerosols were found to broaden the droplet spectrum, and cloud dynamic perturbations may lead to an underestimation of the aerosol dispersion effect. This result is helpful for reducing the dispersion effect uncertainty and may benefit cloud parameterizations in global climate models to more accurately assess the indirect aerosol effect.

Additionally, entrainment plays a critical role in the formation and evolution of clouds and the change in droplet spectrum, as well as aerosol indirect effects (Chen et al., 2014, 2015; Andersen and Cermak, 2015). However, it remains unclear whether the entrainment-mixing mechanism is predominantly homogeneous, inhomogeneous, or in between (Andrejczuk et al., 2009; Lehmann et al., 2009). To our knowledge, little attention has been given to the entrainment-mixing mechanism obtained during VOCALS. Using cloud observations obtained from other aircraft (G-1) during VOCALS, Yum et al. (2015) showed that both homogeneous and inhomogeneous mixing were found in their analysis and attributed it to method uncertainties. We used a completely different method to re-examine the entrainment-mixing mechanism near the stratocumulus top. As stated by Gerber et al. (2005), in marine stratocumulus clouds, entrainment occurs when the LWC begins to decrease from the bottom of the cloud. In this manuscript, entrainment and non-entrainment zones are thus defined as the regions within 20 m above and below the maximal LWC height, respectively. A comparative analysis of the difference in cloud microphysics between the two zones suggests that the entrainment-mixing mechanism is predominantly extreme inhomogeneous in the stratocumulus clouds during VOCALS. The impacts of entrainment on cloud microphysics are also investigated. Previous studies have noted that applying different assumptions to the entrainment-mixing mechanism would have a significant impact on cloud albedo (Grabowski, 2006; Chosson et al., 2007; Slawinska et al., 2008). Therefore, our results provide insights to improve the understanding of entrainment mixing in stratocumulus clouds and the assessment of aerosol indirect effects and cloud radiative forcing.

According to the reviewer's suggestions, we have added the corresponding discussions to the abstract and introduction to ensure clarity of the novelty and contribution of this study, and we removed the paper section layout discussion from the text.

2) I have a couple of concerns about the treatment of the interstial aerosol. First, the PCASP-100 misses the large fraction of sub-0.1-um aerosols that are unlikely to act as CCN and would therefore remain as interstitial aerosols. Are there data from in-cabin particle counters sampling on an aerosol inlet that could fill in this major gap? Inlet shatter may be more of an issue here, but the sub-cloud measurements would be a good place to quantify the fraction of aerosol number that is sub- and super-0.1-um diameter. It is likely that many (or most) of the interstitial aerosol number is not being captured here.

We agree that the concentration of interstitial aerosols ($> 0.1$ μm) measured by PCASP-100 is less than the concentration of all un-activated aerosols, i.e., the interstitial aerosols with diameters smaller than 0.1 μm are not captured here. However, because in-cloud sampling of this part of the aerosols is problematic due to cloud droplet shatter (Hudson and Frisbie, 1991; Clarke et al., 1997; Weber et al., 1998; Kleinman et al., 2012), we do not include it into interstitial aerosols in this manuscript. As suggested by Kleinman et al. (2012), cloud droplet shatter can create a large number of spurious small particles ($\sim$ 50 nm), leading to a serious overestimation of interstitial aerosol concentration. Additionally, other previous studies also observed extremely high concentrations of these small particles in-cloud (e.g., $10^3$ to more than $10^4$ particles cm$^{-3}$ smaller than 50 nm diameter) and attributed these concentrations to droplet shatter (Hudson and Frisbie, 1991; Clarke et al., 1997; Weber et al., 1998). However, the shatter contribution to the total in-cloud aerosols is minor when the diameter is greater than 0.1-0.15 μm (Kleinman et al., 2012). Therefore, treating aerosols larger than 0.1 μm as interstitial aerosols can avoid the interference of cloud droplet shatter to a large extent, which is precisely what we did in this manuscript.

According to the reviewer's suggestions, we calculated the ratio of the sub-0.1-μm aerosol concentration to the super-0.1-μm aerosol concentration during flights on Oct. 18, where sub-0.1-μm aerosol concentration is derived from the concentration measured by CPC (size range: $> 15$ nm) minus that measured by PCASP-100 (size range: 0.1-2.0 μm), and the super-0.1-μm aerosol concentration is obtained from the concentration measured by PCASP-100. It is found that the average ratio for in-cloud (2.99) is significantly higher than that for sub-cloud (0.31), which further confirms the contribution of droplet shatter to the spurious increase in small aerosols in clouds. Notably, the average ratio for sub-cloud is only 0.31, indicating that some aerosols could be missed by PCASP-100, but these aerosols are not the majority. Furthermore, as the reviewer stated, most aerosols smaller than 0.1 μm are unlikely to act as CCN and thus affect cloud properties. Therefore, due to the weak connection with CCN, these aerosols that are missed by PCASP-100 may not be important when exploring the impact of aerosols on clouds. For example, the same treatment of interstitial aerosols as in this manuscript has also been employed by other studies (Kleinman et al., 2012) to examine the aerosol effect on clouds.

3) The use of effective diameters to characterize the interstial aerosol (Di) and cloud droplet sizes (De) doesn't make sense to me as this paper is largely focused on number concentrations and number size distributions. While the effective diameter is relevant for remote sensing measurements, there are no remote sensing data presented in this paper. The authors should instead use geometric mean diameters to describe these aerosol populations and better convey the aerosol and cloud diameters relevant for the number distributions.

Thanks for the comments. Our focus in this manuscript is to investigate the effect of aerosols on clouds and hence provide a reference for the assessment of aerosol indirect radiative forcing. The climatic effects of aerosols and clouds are usually estimated based on the radiatively important effective diameters rather than the geometric mean diameters. For this reason, many aircraft-based studies on aerosol-cloud interactions also used effective diameters to represent the sizes of aerosols and cloud droplets (Peng et al., 2002; Zhang et al., 2011; Vogelmann et al., 2012; Yang et al., 2019; Zhao et al., 2018, 2019), although there was no comparison with the remote sensing measurements. Additionally, effective diameters are often used to quantify the extent of dilution of the cloud caused by the entrainment-mixing process (Pontikis et al., 1993; Gerber et al., 2008, 2013; 2016). For the above reasons, we used effective diameters instead of geometric mean diameters in this manuscript.

We agree with the reviewer that the geometric mean diameter is more relevant for the number size distributions. Thus, we compared the effective diameters with the geometric mean diameters of aerosols and cloud droplets, respectively (Figure R1), and found that there is a good correlation between them. This correlation implies that using the two different diameters would not influence our conclusions.

[Figure]

Figure R1. Correlation between geometric mean diameters and effective diameters of (a) aerosols and (b) cloud droplets, respectively. Colors represent the frequency (units: %).

4) In a number of instances associations between sub-cloud and in-cloud variables are misinterpreted to suggest causal relationships that are inconsistent with our understanding of cloud physics. For exampele, on Lines 13-15, it is stated "Our analysis suggest (sic) that the increase in liquid water content (LWC) is mainly contributed by cloud droplet number concentration (Nd) instead of effective radius of cloud droplets in the polluted case, in which more droplets form with smaller size, while the opposite is true in the clean case." On Lines 142-144, it is stated: "This may imply that the increase of LWC induced by sub-CCN is mainly caused by increasing Nd instead of Re. Fig. 3d indicates a positive correlation between cloud depth and sub-CCN..." These statements are either misleading or just not correct. LWC is known to be driven by changes in environmental conditions (i.e., profiles of temperature and total water content as well as entrainment mixing); microphysics are not a primary driver. Similarly, changes in these environmental conditions will also change the cloud base altitude (and hence cloud depth if the cloud top is driven by a constant

inversion height). There is a causal link between sub-CCN and droplet number, while the in cloud supersaturation (again driven by environmental conditions) can also affect Nd. What the analysis shown in Figure 3 does suggest is that there is a correlation between higher sub-CCN loadings and wetter (or colder) environmental conditions, which should be discussed. The old axiom that correlation does not imply causation certainly holds here. These conclusions (on the lines cited above and elsewhere in the manuscript) need to be either revised or removed from the manuscript.

Thanks for the comments. We agree with the reviewer that both environmental conditions and microphysics (i.e., aerosol effects) can affect cloud properties, and usually, the former is the primary driver. On this point, we distinguished between the flights of the typical mixed BL and others to ensure relatively similar meteorological conditions (similar inversion heights, and the jump of potential temperature and total water mixing ratio across the inversion). In addition, the in-cloud dynamics (i.e., vertical velocity) for the 16 non-drizzling flights were also compared (Figure 9 and Table 2). The result indicated that the in-cloud dynamic differences between the typical well-mixed boundary flights is very small, which confirms the assumption of similar meteorological conditions. Therefore, the interference of environmental conditions on the relationships between the cloud and sub-CCN shown in Figure 3 would be minimal.

LWC is a function of both the number (Nd) and size (Re) of cloud droplets, i.e., both Nd and Re contribute to the change in LWC. The objective of the analysis on the relationships between LWC and Nd/Re (Lines 13-15; Figure 4) is to understand which contributed more to the growth of LWC under polluted and clean conditions, i.e., cloud formation under different aerosol loadings. Our analysis shows that the low aerosol concentrations in the clean case inhibit the increase in Nd with LWC, which promotes the rapid increase in Re with LWC. In contrast, there are enough particles that may potentially be activated into cloud droplets under polluted conditions; thus, Nd increases rapidly with LWC, while the increase in Re is suppressed. We agree with the reviewer that the sentences on lines 13-15 and lines 142-144 might be misunderstood, and thus, these sentences have been removed from revised manuscript as suggested.

Minor Comments:

1) On Line 76, it is stated that the PCASP-100 measures the aerosol dry diameter. How was this accomplished? Was some sort of unique inlet heater or dryer used to dry the aerosol? While there will be some ram heating effects that will lower the relative humidity in the PCASP-100 optics region, I don't think that this would be enough to say that the aerosol size is dry.

Thanks for this comment. The relative humidity of the air sampled by PCASP can be reduced to 40 % by using deicing heaters, which increases the temperature by approximately 10 to 20 °C (Strapp et al., 1992; Hallar et al., 2006; Snider and Petters, 2008). Cabin heat also contributed to the drying of aerosols measured inside the aircraft. We agree with the reviewer that this may be not enough to say the diameter is dry, although the relative humidity is already very low. Thus, we have revised "dry diameter" to "diameter" on line 104.

2) Too many significant figures reported on Line 158. Reporting aerosol concentration as integer

values would be appropriate.

Thanks for the suggestion. In the revised manuscript, the number concentrations of aerosols and cloud droplets were reported as integer values throughout the paper.

3) The manuscript would benefit from some additional proofreading to improve grammatical or typo errors.

We have carefully proofread the manuscript and asked a native English speaker to read and edit the language of the manuscript.

[Figure]

**Fig. 8.** Relationships between relative dispersion (ε) and $N_d$ during all 16 non-drizzling flights, in which the colors  **represents** in-cloud vertical velocities (m s[-1]).

760

[Figure]

[Figure]

**Fig. 9.** Probability distribution function (units: %) of vertical velocity (*w*) for 16 non-drizzling flights. Black symbols are mean values of *w*, and error bars through these symbols indicate the standard deviation. Circles are the cases with typical well-mixed BL, and crosses represents the other cases.

[Figure]

**Fig. 10.** Relative dispersion (*ε*) as a function of (a) $N_d$ and (b) sub-cloud CCN concentrations (*SS*=0.2%) for all flights. The error bars through these symbols indicate the standard deviation. Red symbols are the cases with typical  well-mixed BL , and blue symbols for others. Red (black) texts are the correlation coefficient and slope for typical well-mixed cases (all cases).

[Figure]

**Fig. 11.** Number size distributions of cloud droplets in the entrainment (yellow) and non-entrainment zones (blue) during all 16 non-drizzling flights.

[Figure]

[Figure]

**Fig. 12. Probability density functions of (a)**  $R_i$ **(μm), (b)** *RH* **(%), (c)** $N_d/(N_d + N_i)$**, and (d)** *ε* **in** the **entrainment (yellow) and non-entrainment zone**s **(blue) during the flight on Oct. 18.**

*Supplement of*

**Exploring aerosol cloud interaction using VOCALS-REx aircraft measurements**

Hailing Jia, Xiaoyan Ma and Yangang Liu

*Correspondence to*: Xiaoyan Ma (xma@nuist.edu.cn)

**Figure List**

Figure S1. Normalized profiles of $N_d$. Values of $Z_N=0$ indicates the cloud base whereas $Z_N=1$ the cloud top. Orange line indicates the average profiles.

Figure S2. (a) $P_{LWC}$ and (b) $P_{Nd}$ as a function of $AF_{ent}/AF_{non-ent}$ for all 16 non-drizzling flights.

**Figure S1**

[Figure]

**Figure S2**

[Figure]